# DINO AS A VON MISES-FISHER MIXTURE MODEL

**Hariprasath Govindarajan**[1,2]  **Per Sidén**[1,2]  **Jacob Roll**[2]  **Fredrik Lindsten**[1]
[1]Linköping University, Sweden  [2] Qualcomm Technologies, Inc.
`{hargov,psiden,jroll}@qti.qualcomm.com,`
`fredrik.lindsten@liu.se`

## ABSTRACT

Self-distillation methods using Siamese networks are popular for self-supervised pre-training. DINO is one such method based on a cross-entropy loss between $K$-dimensional probability vectors, obtained by applying a softmax function to the dot product between representations and learnt prototypes. Given the fact that the learned representations are $L^2$-normalized, we show that DINO and its derivatives, such as iBOT, can be interpreted as a mixture model of von Mises-Fisher components. With this interpretation, DINO assumes equal precision for all components when the prototypes are also $L^2$-normalized. Using this insight we propose DINO-vMF, that adds appropriate normalization constants when computing the cluster assignment probabilities. Unlike DINO, DINO-vMF is stable also for the larger ViT-Base model with unnormalized prototypes. We show that the added flexibility of the mixture model is beneficial in terms of better image representations. The DINO-vMF pre-trained model consistently performs better than DINO on a range of downstream tasks. We obtain similar improvements for iBOT-vMF vs iBOT and thereby show the relevance of our proposed modification also for other methods derived from DINO.

## 1 INTRODUCTION

Self-supervised learning (SSL) is an effective approach for pre-training models on large unlabeled datasets. The main objective of SSL pre-training is to learn representations that are transferable to a range of, so called, downstream tasks. Early SSL methods achieved this through handcrafted pretext tasks that act as inductive biases in the representation learning process (Komodakis & Gidaris, 2018; Noroozi & Favaro, 2016; Kim et al., 2018; Doersch et al., 2015; Larsson et al., 2016; Zhang et al., 2016). Contrastive methods (Tian et al., 2020; Wu et al., 2018), using supervisory signals in the form of augmentation invariances and instance discrimination, have produced strong performance benchmarks. In practice, contrastive methods require large batch sizes (Chen et al., 2020a) or specialized techniques like memory banks (He et al., 2020; Misra & Maaten, 2020) to achieve the best performance. Self-distillation methods based on the Mean Teacher (Tarvainen & Valpola, 2017) framework are effective representation learners that do not require such large batch sizes for pre-training. A trivial solution where the network learns to output the same representation irrespective of the input is known as representation collapse. The negative samples in contrastive learning prevent representation collapse. In the absence of negative samples, self-distillation methods use explicit approaches to avoid collapse, such as asymmetric model architecture (Grill et al., 2020; Chen & He, 2021) and whitening (Ermolov et al., 2021; Zbontar et al., 2021).

Transformers (Vaswani et al., 2017), originally introduced in NLP, have emerged as a strong model architecture for vision tasks as well (Dosovitskiy et al., 2020; Liu et al., 2021). Current state-of-the-art SSL methods leverage the highly flexible Vision Transformers (ViTs) (Caron et al., 2021; Bao et al., 2021; Li et al., 2021; Zhou et al., 2021; Xie et al., 2021; Chen et al., 2021). DINO (Caron et al., 2021) is a non-contrastive SSL method that is effective for pre-training ViT models. Interestingly, ViTs pre-trained using DINO outperformed ResNets (He et al., 2016) by a significant margin at kNN classification based on learned representations. DINO is an influential SSL method with several state-of-the-art derivatives: MSN (Assran et al., 2022) adapts DINO to produce strong few-shot performance with enhanced training efficiency; iBOT (Zhou et al., 2021) extends DINO by adding an additional masked image modeling task; EsViT extends DINO by adding patch-level tasks that also

use a DINO-like formulation. These methods are all trained using the Mean Teacher framework by learning to produce consistent outputs in the probability simplex. The networks output softmax-logit scores based on an inner product between a learned representation and a set of prototypes.

We provide a better understanding of DINO, and its derivatives, by taking a closer look at its inner-product formulation. We interpret DINO as a von Mises-Fisher mixture model under certain assumptions. Based on this interpretation, we propose DINO-vMF, as a modified version of DINO, that adds flexibility in the learned latent space while keeping the training stable. DINO-vMF pre-training consistently improves performance on similar downstream tasks as DINO. We also show that the larger ViT models achieve significantly improved few-shot classification performance with our pre-training. By incorporating our vMF modification in iBOT, we achieve significantly improved performance that suggests that our method is applicable to DINO-derived methods as well.

## 2 DINO

The self-distillation learning framework in DINO considers a teacher network $g_{\theta_t}$ and a student network $g_{\theta_s}$, with parameters $\theta_t$ and $\theta_s$, respectively. In DINO, the student network is formulated to predict a vector in the $(K-1)$-dimensional probability simplex using a softmax function. The student probability distribution is obtained as follows:

$$P_s^{(k)}(\boldsymbol{x}) \overset{k}{\propto} \exp g_{\theta_s}^{(k)}(\boldsymbol{x}) \tag{1}$$

where $\overset{k}{\propto}$ indicates that the right-hand side is normalized w.r.t. the index $k$ (i.e. the equation above corresponds to a softmax). The teacher probability distribution $P_t(\boldsymbol{x})$ is computed analogously.

Given an unlabeled image dataset $\mathcal{I}$, consider uniform samples $\boldsymbol{x} \sim \mathcal{I}$ and two random augmentations $A_s \sim \mathcal{A}_s, A_t \sim \mathcal{A}_t$. By applying these augmentations, we get two views $\boldsymbol{x}_s = A_s(\boldsymbol{x})$ and $\boldsymbol{x}_t = A_t(\boldsymbol{x})$. The student network is trained using gradient updates to produce outputs $P_s(\boldsymbol{x}_s)$ that are consistent with those of the teacher network $P_t(\boldsymbol{x}_t)$ by minimizing a cross-entropy loss given by: $\min_{\theta_s} \sum_{k=1}^{K} -P_t^{(k)}(\boldsymbol{x}_t) \log P_s^{(k)}(\boldsymbol{x}_s)$. The teacher network parameters $\theta_t$ are only updated as an exponential moving average (EMA) of $\theta_s$.

SSL methods based on Siamese networks (Bromley et al., 1993) face the representation collapse problem, where the network learns to produce the same output irrespective of the input. One approach to address this is by introducing an asymmetry between the teacher and student networks. DINO uses the same model architecture for the student and teacher networks but instead shows that adding asymmetry through centering and sharpening operations is sufficient to avoid collapse. The targets produced by the teacher network are centered to remove bias towards a cluster and sharpened using a temperature $\tau_t$ as $g_{\theta_t}(\boldsymbol{x}_t) \leftarrow (g_{\theta_t}(\boldsymbol{x}_t) - \boldsymbol{c})/\tau_t$. On the other hand, the student outputs are only sharpened as $g_{\theta_s}(\boldsymbol{x}_s) \leftarrow g_{\theta_s}(\boldsymbol{x}_s)/\tau_s$. The centering operation prevents one of the probability components from dominating but the solution could collapse to a uniform distribution instead. This is avoided by the sharpening operation, where the teacher uses a lower temperature value than the student, $\tau_t < \tau_s = 0.1$. The overall schematic of DINO is illustrated in Figure 1a.

## 3 METHOD

### 3.1 DINO: A CLOSER LOOK AT THE FINAL LAYER

The teacher and student networks are designed by combining a backbone network that can be transferred to downstream tasks and a prediction head that is specific to the SSL pre-training. We take a closer look at the prediction head as shown in Figure 1b, which is found to be important to achieve good performance in ablation experiments (Caron et al., 2021). The weight-normalization in the last linear layer (Salimans & Kingma, 2016) refers to a reparameterization of the weights $W$ as $\boldsymbol{w}^{(k)} = g^{(k)} \boldsymbol{v}^{(k)}$ where $\boldsymbol{w}^{(k)}$ is the $k$th column of $W$, $g^{(k)}$ is a scalar magnitude, and $\boldsymbol{v}^{(k)}$ is a unit vector. Optionally, the weights are $L^2$-normalized by fixing $\boldsymbol{g} = \boldsymbol{1}$.

Whether or not to $L^2$-normalize the prototypes is a subtle, and in our opinion *ad hoc*, design choice which is not discussed enough in the literature. Most prior clustering-based SSL methods do use normalized prototypes (Caron et al., 2018; 2019; 2020; Li et al., 2020; Asano et al., 2020), and this

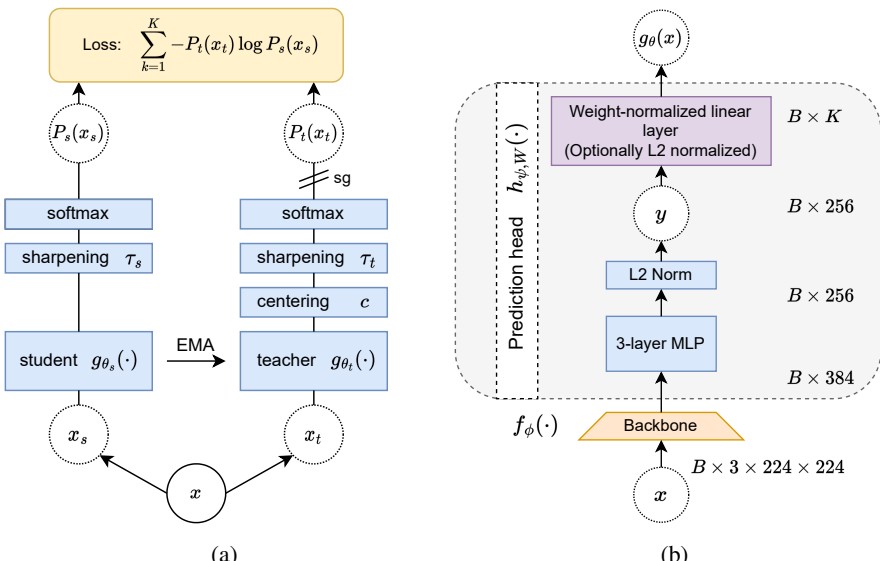

Figure 1: **Overview of DINO.** (a): High-level architecture of DINO; (b): A closer look at the networks $g_\theta$, modeled as a combination of a backbone $f_\phi$ and a prediction head $h_{\psi,W}$, where $\theta = \{\phi, \psi, W\}$. The prediction head contains 3 MLP layers, an $L^2$-normalization bottleneck and a weight-normalized (Salimans & Kingma, 2016) linear layer. The weights of the weight-normalized linear layer are $L^2$-normalized in the larger ViT-Base models to ensure stable training.

is not mentioned as a design choice in the paper by Caron et al. (2021), nor in their supplementary material. However, from the public code repository of DINO[1], it can be noted that *not* $L^2$-normalizing $W$ leads to a performance boost for ViT-Small models (these are the results reported in their paper). However, the larger ViT-Base models require the weights to be $L^2$-normalized since the training is unstable otherwise. Furthermore, comparing the ViT-Small and ViT-Base models at patch size 8, surprisingly the ViT-Small model achieves better kNN top-1 validation accuracy on ImageNet (Caron et al., 2021, Table 2). This suggests that $L^2$-normalization is an important design choice and that the representations learned by the ViT-Base model (i.e. with $L^2$-normalization) are sub-optimal and has room to improve. In the next section we provide a better understanding of this from a mixture model perspective and investigate how we can better leverage prototypes that are not $L^2$-normalized.

## 3.2 DINO as a Mixture Model

DINO learns to predict a soft distribution over $K$ components. Let the intermediate output in the prediction head after the $L^2$-normalization of the representation[2] be denoted as $y$; see Figure 1b. Then, DINO can be interpreted as performing clustering in the latent space of $y$. Specifically, since $y$ is a unit vector, DINO performs clustering on the unit hypersphere.

Clustering is closely related to mixture modeling. On the unit hypersphere, a mixture model can be defined using von Mises-Fisher (vMF) components. For a random $p$-dimensional unit vector $y$, the vMF probability density function is given by $f(y; \mu, \kappa) = C_p(\kappa) \exp(\kappa \mu^T y)$, where $\mu$ is a mean vector with $\|\mu\| = 1$, $\kappa$ is a scalar concentration parameter that measures isotropic precision, and $C_p(\kappa)$ is a normalizing constant defined as: $C_p(\kappa) = \kappa^{p/2-1}/\left[(2\pi)^{p/2} I_{p/2-1}(\kappa)\right]$, where $I_\nu$ denotes the modified Bessel function of the first kind and order $\nu$. Assuming a mixture model containing $K$ vMF components with mixture component ratios $\pi^{(k)}$, the probability of assigning a sample $y_i$ to a mixture component $k$ (also known as the responsibility of cluster $k$) is given by:

$$r_i^{(k)} \overset{k}{\propto} \pi^{(k)} \Pr(y_i|z_i = k) = \pi^{(k)} C_p(\kappa^{(k)}) \exp\left[\langle \kappa^{(k)} \mu^{(k)}, y_i \rangle\right] \tag{2}$$

where $\langle u, v \rangle = u^T v$ denotes inner product.

---

[1]https://github.com/facebookresearch/dino

[2]Note that this is different from the $L^2$-normalization of the prototypes.

In DINO, the probability distributions for the student and the teacher can be rewritten as:

$$P_s^{(k)}(\boldsymbol{x}_s) \overset{k}{\propto} \exp\left[\langle \boldsymbol{w}_s^{(k)}, \boldsymbol{y}_s \rangle / \tau_s \right], \tag{3}$$

$$P_t^{(k)}(\boldsymbol{x}_t) \overset{k}{\propto} \exp\left[\left(\langle \boldsymbol{w}_t^{(k)}, \boldsymbol{y}_t \rangle - c^{(k)}\right) / \tau_t \right] = \exp\left[-c^{(k)}/\tau_t\right] \exp\left[\langle \boldsymbol{w}_t^{(k)}, \boldsymbol{y}_t \rangle / \tau_t \right]. \tag{4}$$

Note that centering is only applied on the teacher outputs and that different sharpening temperatures are used for the teacher and the student. By comparing Eq. (3) with Eq. (2), we observe that the DINO formulation resembles that of a mixture model, given some assumptions. We explain these assumptions below to establish a more concrete connection. This interpretation also applies to other recent works that use a similar inner-product-based prototype formulation (Caron et al., 2020; Assran et al., 2022; Li et al., 2021; 2020).

The exponential term in DINO from Eq. (3) corresponds to the unnormalized probability density of a vMF distribution if we identify $\kappa^{(k)} = \|\boldsymbol{w}^{(k)}\|/\tau$ and $\boldsymbol{\mu}^{(k)} = \boldsymbol{w}^{(k)}/\|\boldsymbol{w}^{(k)}\|$. Similar to prior clustering-based SSL works, DINO avoids collapse by encouraging a uniform distribution of the data over the prototypes, which explains the absence of $\pi^{(k)}$ in Eq. (3). In the teacher model, Eq. (4), we obtain an additional term through the centering operation which further encourages uniformity. We discuss more about the centering operation in section 3.4.

Following the mixture model interpretation, we note that a missing term in the DINO formulation is the normalization constant, $C_p(\kappa^{(k)})$. However, this is inconsequential when the prototypes $\boldsymbol{w}^{(k)}$ are $L^2$-normalized as it will result in constant $\kappa^{(k)} = 1/\tau$. Hence, the constant $C_p(\kappa^{(k)})$ would vanish in the softmax. However, a constant $\kappa^{(k)}$ implies the assumption that all clusters should have a similar isotropic precision. This constraint reduces the flexibility of the mixture model, and this lack of flexibility in the latent space translates to the backbone mapping as well. Indeed, as discussed above, unnormalized prototypes help in improving the performance achieved by DINO (for ViT-Small models) by enabling better representations (Caron et al., 2021). However, larger ViT-Base models are affected by training instabilities when trained with unnormalized prototypes. We hypothesise that this is due to the "missing" normalization constant and propose to modify DINO by including appropriate normalization according to a vMF mixture model.

### 3.3 NORMALIZING VON MISES-FISHER (VMF) COMPONENTS

In DINO, a large $\|\boldsymbol{w}^{(k)}\|$ scales the logit scores of a mixture component proportionally. In a properly normalized formulation, a large $\|\boldsymbol{w}^{(k)}\|$ instead scales the $\kappa^{(k)}$ parameter proportionally and results in a sharper vMF distribution for the component. Hence, the model cannot naively increase $\|\boldsymbol{w}^{(k)}\|$ to increase the responsibility of a component to a data sample—it also needs to map the data samples close to the component's prototype. If the images assigned to a component can be consistently mapped close to its prototype in the latent space, only then it is beneficial to increase $\kappa^{(k)}$.

The vMF normalization constant $C_p(\kappa)$ includes the modified Bessel function of the first kind and order $\nu$, denoted as $I_\nu(\kappa)$. It is easy to obtain $\kappa = \|\boldsymbol{w}\|/\tau$, but since $I_\nu(\kappa)$ is a complicated function, we propose to use a differentiable approximation thereof. Let $\nu = p/2 - 1$ and $\kappa = \nu r$. Then, $I_\nu$ can be approximated by the following uniform expansion for large $\nu$ (DLMF, Eq. 10.41.3):

$$I_\nu(\nu r) \sim \frac{e^{\nu\eta}}{(2\pi\nu)^{1/2}(1+r^2)^{1/4}} \sum_{i=0}^{\infty} \frac{U_i(s)}{\nu^i}, \quad \eta = (1+r^2)^{1/2} + \log\frac{r}{1+(1+r^2)^{1/2}} \tag{5}$$

where $\sim$ denotes a Poincaré asymptotic expansion w.r.t. $\nu$, $s = (1+r^2)^{-1/2}$ and polynomials $U_i(s)$. Here, the bottleneck dimension in the prediction head is $p = 256$, and this results in $\nu = 127$, which is large in this context. Empirically, evaluating only the first term in the sum, $U_0(s) = 1$ seems to be sufficient, see Appendix A.2 for details. With this approximation, we compute $C_p(\kappa^{(k)})$ for the $k$-th component and modify the logit scores for component $k$ for both teacher and student from $(\langle \boldsymbol{w}^{(k)}, \boldsymbol{y} \rangle)/\tau$ to $(\langle \boldsymbol{w}^{(k)}, \boldsymbol{y} \rangle)/\tau + \log C_p(\kappa^{(k)})$. That is, we add $\log C_p(\kappa^{(k)})$ to the logit scores in (3) and (4) to obtain the following expressions for DINO-vMF:

$$P_s(\boldsymbol{x}_s)^{(k)} \overset{k}{\propto} \exp\left[\langle \boldsymbol{w}_s^{(k)}, \boldsymbol{y}_s \rangle / \tau_s + \log C_p\left(\kappa_s^{(k)}\right)\right], \tag{6}$$

$$P_t(\boldsymbol{x}_t)^{(k)} \overset{k}{\propto} \exp\left[-c^{(k)}\right] \exp\left[\langle \boldsymbol{w}_t^{(k)}, \boldsymbol{y}_t \rangle / \tau_t + \log C_p\left(\kappa_t^{(k)}\right)\right]. \tag{7}$$

### 3.4 Avoiding Collapse

The probability distributions are computed based on the teacher and student network outputs using a softmax function. With the motivation of avoiding collapse, *centering* and *sharpening* operations are applied to the network outputs to obtain the logit scores for the probability distribution. The centering parameter $c$ in DINO is computed as an EMA of the teacher network outputs with a forgetting parameter $m \lesssim 1$ as follows:

$$c^{(k)} \leftarrow mc^{(k)} + (1-m)\frac{1}{B}\sum_{b=1}^{B}\langle \boldsymbol{w}_t^{(k)}, \boldsymbol{y}_{t,b}\rangle \tag{8}$$

However, when centering is done in the logit space as in (8) with our modified logit scores $(\langle \boldsymbol{w}_t^{(k)}, \boldsymbol{y}_t\rangle)/\tau + \log C_p(\kappa_t^{(k)})$, an EMA of the log normalization constant will be added to $c^{(k)}$. When the centering operation is performed on the teacher outputs, the impact of the log normalization constant is dampened. To avoid this, we propose to instead compute the probability distributions for the images in the batch, and then average the probability distributions instead. This is also closer to how mixture proportions $\pi^{(k)}$ are computed in the M-step of the EM algorithm for a mixture model, although the estimated probabilities play a different role in DINO; see Appendix A.4 for a discussion. Our proposal for computing the centering parameter $c$ is thus:

$$c^{(k)} \leftarrow mc^{(k)} + (1-m)\log\left[\frac{1}{B}\sum_{b=1}^{B}\left(\text{softmax}\left(\boldsymbol{w}_t^T \boldsymbol{y}_{t,b}/\tau_t\right)\right)^{(k)}\right]. \tag{9}$$

## 4 Related Work

**Clustering-based and prototypical methods:** SSL has advanced significantly in recent times and a line of work based on clustering have also evolved. Caron et al. (2018) proposed a predictive task using "pseudo-labels" generated by K-means clustering in the representation space. Following its success, several methods based on clustering have been proposed (Caron et al., 2020; Asano et al., 2020; Caron et al., 2019; Li et al., 2020; Zhuang et al., 2019; Assran et al., 2022; 2021). These methods learn a set of prototype vectors or cluster centroids that correspond with the clusters. When combined with ViTs and self-distillation, Caron et al. (2021) demonstrated that the learned representations possess useful properties – strong nearest neighbor embeddings and rich spatial semantic features. While the DINO formulation was only based on the [CLS] representation, several recent works have extended DINO to utilize patch-level representations in the objective formulation (Li et al., 2021; Zhou et al., 2021). A common aspect among most recent methods is that they utilize $L^2$-normalized prototypes. Our work is based on the DINO formulation. By incorporating our work in iBOT, we show that our insights can prove beneficial to other methods as well. Our method differs from prior works by proposing a way to remove the $L^2$-normalization constraint on the prototypes and hence, enable a flexible mixture model in the representation space. Assran et al. (2022) uses an entropy regularization to avoid collapse and our centering computation shares similarity with it, in terms of computing intra-batch averages of probability distributions. However, we only average teacher probability distributions to compute the centering parameter instead of applying an explicit regularization of the student outputs.

**Mixture of von Mises-Fisher distributions:** Banerjee et al. (2005) proposed a mixture of vMF (movMF) distribution to perform clustering on the unit hypersphere. The EM updates are often done using an approximate reparameterization of the vMF precision parameter $\kappa$ (Banerjee et al., 2005; Barbaro & Rossi, 2021). Instead, we learn $\kappa$ directly through the prototype magnitudes. Some image segmentation methods have used movMF formulations (Yang et al., 2020; Hwang et al., 2019) but assume a fixed $\kappa$ for all mixture components to simplify computations. Taghia et al. (2014); Gopal & Yang (2014) use a complete Bayesian approach based on variational inference to model data as a movMF. Recent works relate the cosine similarity formulation commonly used in contrastive SSL to a vMF density (Shwartz-Ziv et al., 2022; Lee et al., 2021; Wang & Isola, 2020). Our mixture model interpretation is closest to Hasnat et al. (2017) that proposes a movMF loss to train a supervised face verification model, but they make similar uniformity assumptions as DINO.

Table 1: ImageNet kNN classification accuracy ablating on the impact of $L^2$-normalization of prototypes, vMF normalization and probability centering. Average over 2 runs are reported. Refer A.6.1 for results of individual runs.

| Normalization | Centering space | |
| --- | --- | --- |
| | logit | probability |
| None | 70.18 † | 70.49 |
| $L^2$ | 69.51 ‡ | 69.86 |
| vMF | 70.21 | **70.87** |

† = default setting for DINO/ViT-S,
‡ = default setting for DINO/ViT-B.

Table 2: ImageNet classification accuracy using linear and kNN classifiers. ([C] = `[CLS]`, [D] = DINO)

| Method | kNN | Lin. [C] | Lin. [D] |
| --- | --- | --- | --- |
| *ViT-Base/16* | | | |
| iBOT | 77.10 | 79.33 | 79.50 |
| iBOT-vMF | **78.66** | **80.20** | **80.27** |
| DINO | 76.11 | 77.88 | 78.17 |
| DINO-vMF | 77.40 | 78.67 | 78.81 |
| BeIT-v2 | – | – | 80.10 |
| MSN | 73.33 | 75.84 | 74.80 |
| *ViT-Small/16* | | | |
| DINO | 74.44 | 76.09 | 76.98 |
| DINO-vMF | 74.74 | 76.19 | 76.97 |
| MSN | 74.86 | **76.20** | 76.62 |
| iBOT | **75.20** | – | **77.90** |

## 5 EXPERIMENTS

We closely follow the training settings for different backbones as specified in the public repository for DINO[3], in order to ensure fair comparison. The models are pre-trained on the ImageNet dataset (Deng et al., 2009) with the adamw optimizer (Loshchilov & Hutter, 2018). Weight decay with a cosine schedule from 0.04 to 0.4 is used. The student temperature $\tau$ is set to 0.1 and the teacher temperature is linearly scaled from 0.04 to 0.07 over some initial epochs (50 epochs for ViT-Small/16 and 30 epochs for ViT-Base/16). We follow the same data augmentations as DINO and BYOL (Grill et al., 2020): color jittering, Gaussian blur and solarization. The multi-crop approach is used with the same definition of global and local crop sizes as DINO. The model trainings are done on a single A100 node, consisting of 8 GPUs. The batch sizes are adapted to fit the node and adjusted based on the model architecture (batch size=64 per GPU for ViT-Base/16 and 128 for ViT-Small/16). The vMF normalization computation does not add any noticeable overheads to standard DINO and iBOT. Due to computational reasons, we train ViT-Small/8 for only 40 epochs (refer A.6.2) and do not consider ViTs larger than the Base variant. For comparison of results, we use the publicly released DINO and MSN[4] pre-trained models available in their respective repositories.

### 5.1 IMAGENET CLASSIFICATION

#### 5.1.1 ABLATION STUDIES

We conducted ablation experiments to study the impact of our proposed modifications to DINO. We consider small-scale experiments by training a ViT-Small/16 backbone for 100 epochs with all other settings maintained similar to the standard training. We report the kNN top-1 classification accuracy on ImageNet in Table 1 by averaging over 2 runs. We observe that the performance is better when the cluster prototypes are not $L^2$-normalized. We find that vMF normalization and probability centering individually lead to performance improvements. Best performance is achieved when both vMF normalization and centering in probability space are used simultaneously. On this basis, we decide to use vMF normalization along with centering in the probability space in all other experiments. Note that maximal improvement is observed when compared to the default setting of DINO/ViT-B.

#### 5.1.2 IMAGENET CLASSIFICATION WITH FULL DATASET

The quality of representations learned by self-supervised models are usually evaluated by linear classification benchmarks on top of a frozen pre-trained backbone model. DINO is also shown to be an efficient nearest neighbor classifier. For linear evaluation, we follow the same protocol as DINO. Additionally, we also evaluate the performance using only the `[CLS]` representation. For

---

[3]https://github.com/facebookresearch/dino
[4]https://github.com/facebookresearch/msn

Table 3: ImageNet few-shot evaluation (as in MSN)

| | ViT-Base/16 | | | | | ViT-Small/16 | | |
| Dataset | DINO | MSN | DINO-vMF | iBOT | iBOT-vMF | DINO | MSN | DINO-vMF |
|---|---|---|---|---|---|---|---|---|
| 1 img/cls | 41.8 | 49.8 | **50.3** | 46.0 | **51.6** | 38.9 | **47.1** | 39.2 |
| 2 imgs/cls | 51.9 | 58.9 | **59.3** | 56.0 | **61.1** | 48.9 | **55.8** | 49.4 |
| 5 imgs/cls | 61.4 | 65.5 | **66.1** | 64.7 | **68.3** | 58.5 | **62.8** | 59.1 |
| 1% imgs | 67.2 | 69.1 | **70.4** | 69.9 | **72.3** | 64.5 | **67.2** | 65.0 |

Table 4: Prototype utilization based on a cosine similarity threshold of 0.9.

| | # unique prototypes | | Largest duplicate prototype set | |
| Architecture | DINO | DINO-vMF | DINO | DINO-vMF |
|---|---|---|---|---|
| ViT-Small/16 | 961 | 1122 | 2087 | 1226 |
| ViT-Base/16 | 775 | 928 | 54877 | 1208 |

kNN evaluation, we consider a straightforward sweep over a range of values for nearest neighbors and generally find $k = 10$ to work best with DINO-vMF models. The accuracy achieved by linear and kNN classifiers on ImageNet are reported in Table 2 (more baselines in A.6.2). With ViT-Base/16, both DINO-vMF and iBOT-vMF clearly perform better than their standard counterparts. With ViT-Small/16, the performance improvement is found to be marginal. MSN (Assran et al., 2022) shows competitive performance on ViT-Small/16 but it is significantly worse on the larger ViT-Base/16.

### 5.1.3 IMAGENET FEW-SHOT EVALUATION

Few-shot classification is an important application area for SSL pre-training. MSN (Assran et al., 2022) pre-training is found to be effective at few-shot classification tasks. We compare our DINO, iBOT, their vMF versions and MSN pre-training using the same evaluation protocol as MSN. A linear classifier is trained on features obtained using a center crop of the image from the frozen pre-trained model. $L^2$-regularization with a constant strength of 0.075 is used. We consider different levels of labelled data availability: 1, 2 and 5 images per class, and 1% of all the training images. We report the top-1 accuracy in Table 3. For the 1, 2 and 5 images per class experiments, we use three different splits and report the mean of the accuracy (refer A.6.4 for standard deviations). With ViT-Base/16 model, we observe a large improvement by adding the vMF modification and even surpassing the performance achieved by MSN (Assran et al., 2022), which specifically targets the few-shot learning problem. In comparison, the ViT-Small/16 model with DINO-vMF pre-training only improves marginally over DINO and is found to be lagging behind MSN by a large margin.

### 5.2 ANALYSIS OF LEARNED vMF MIXTURE MODEL

In this section, we take a deeper look at the mixture modeling task, which acts as the supervisory signal to train the self-supervised model.

**Prototype utilization:** Inspecting the prototype vectors learned by DINO, we observed that many of them were almost identical. We consider two prototypes $w^{(i)}$ and $w^{(j)}$ to be duplicates if their cosine similarity is greater than a threshold value. We identify unique prototypes by removing such duplicates. Recall that DINO uses a large number of prototypes set to 65536 for all models. Assran et al. (2022) does not observe a benefit in using more than 1024 prototypes, but fewer prototypes lead to worse performance. In Table 4, we show the number of unique prototypes and the size of the largest set of duplicate prototypes based on a cosine similarity threshold of 0.9 (in A.3, we show that our observations hold at other threshold choices as well). Interestingly, when pre-trained with DINO, the ViT-Base model produced one large duplicate set containing $\approx 83\%$ of all the prototypes. Based on highest probability, no data samples are assigned to the corresponding mixture components. We refer to this as the *void prototype set*. Interestingly, the void prototype has a cosine similarity of exactly $-1$ with the mean of $y$ over all the training data. This means that the void prototype vector

points in a direction exactly opposite to the data mean, $\bar{\boldsymbol{y}}$. This occurs because the model minimizes the logit scores of unused prototypes i.e. void prototypes, by moving them as far away as possible from the data representations to minimize their effect on the loss. With DINO-vMF pre-training, we do not observe that any such void prototype set is formed and more unique prototypes are utilized than standard DINO pre-training. Utilizing more prototypes increases the difficulty of the SSL task and we expect this to result in better image representations.

**Interpreting learnt vMF precision:** In the DINO-vMF formulation, the $L^2$-norm of the prototypes is directly related to the vMF precision as $\kappa = \|\boldsymbol{w}^{(k)}\|/\tau$. We investigate if the precision values can be indicative of image classification difficulty by evaluating downstream performance. We associate an image with the component $k$ that has the maximum logit score under the DINO(-vMF) formulation. For each model, we divide the data based on percentile ranges of the precision values of the associated components. In Figure 2, we show the kNN top-1 validation accuracy on ImageNet for subsets of data restricted to lie in the different precision percentile ranges. For DINO-vMF, we observe that the kNN classification accuracy is increasing with increasing precision values. That is, data points associated with higher precision components appear to be easier to classify, and vice versa. The prototype magnitudes learned by DINO (only in ViT-Small/16) do not exhibit such a clear association.

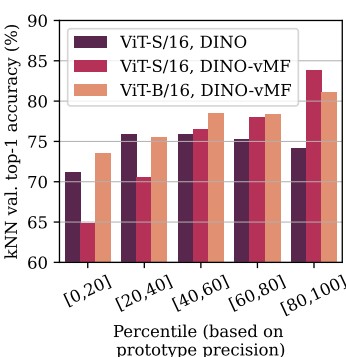

Figure 2: kNN accuracy for data sorted based on percentile ranges of associated $\|\boldsymbol{w}^{(k)}\|$.

## 5.3 Transfer Learning

An important benefit of SSL is that the learned model can be transferred to downstream tasks to achieve competitive performance with limited computational cost or limited training data. The tasks we consider are: linear classification on small datasets, image retrieval and video object segmentation.

**Linear classification on small datasets:** We conduct linear classification experiments on a suite of datasets trained on features extracted from a frozen pre-trained model. The implementation details for this experiment are explained in A.6.3. In Table 5, we report the linear classification accuracy on the test or validation dataset, depending on which is publicly available. First, we observe that the DINO-vMF model performs on par or better than the standard DINO model on most of the datasets with both ViT-Small/16 and ViT-Base/16 architectures. The iBOT-vMF model consistently performs best for most datasets. We also observe that the MSN model performs worse than both the DINO and DINO-vMF models on all datasets. Additional fine-tuning results are given in A.6.5.

**Image Retrieval:** We consider the face-blurred versions (v1.0) of Oxford (Philbin et al., 2007) and Paris datasets (Philbin et al., 2008) and use the medium (M) and hard (H) data splits. We perform nearest neighbour based image retrieval using ImageNet pre-trained frozen features and report the mean average precision (mAP). We follow similar evaluation protocol to DINO but we use newer

Table 5: Linear classification accuracy when transferred to other datasets

| Method | Acft. | Cal101 | C10 | C100 | DTD | Flwrs. | Food | Pets | SUN | Avg. |
|---|---|---|---|---|---|---|---|---|---|---|
| *ViT-Base/16* | | | | | | | | | | |
| DINO | 57.7 | 94.5 | 96.9 | 86.3 | **74.8** | **96.0** | 81.6 | 93.9 | 68.5 | 83.4 |
| DINO-vMF | 57.3 | 94.5 | 97.1 | 86.3 | **74.8** | 95.7 | 82.5 | **94.6** | 68.7 | 83.5 |
| iBOT | 55.9 | 94.7 | 97.6 | 87.4 | 74.4 | 95.4 | 82.7 | 93.8 | 69.4 | 83.5 |
| iBOT-vMF | **58.1** | **95.5** | **98.0** | **88.0** | 74.7 | 94.8 | **83.6** | 93.9 | **70.2** | **84.1** |
| MSN | 51.0 | 92.8 | 96.9 | 85.3 | 73.7 | 92.8 | 80.0 | 93.9 | 66.8 | 81.5 |
| *ViT-Small/16* | | | | | | | | | | |
| DINO | 53.8 | 93.1 | **96.2** | 83.5 | **74.7** | 94.7 | 79.7 | 93.8 | **66.8** | 81.8 |
| DINO-vMF | **56.0** | **93.7** | 96.0 | **83.9** | 74.1 | **95.0** | **80.1** | 93.9 | 66.6 | **82.1** |
| MSN | 51.6 | 93.1 | 95.9 | 82.9 | 72.0 | 93.3 | 77.8 | 92.8 | 65.5 | 80.5 |

Table 6: Image retrieval performance

|  | RPar | | ROx | |
| --- | --- | --- | --- | --- |
| Pretrain | M | H | M | H |
| *ViT-Base/16* | | | | |
| DINO | 63.7 | 35.8 | 34.3 | 10.8 |
| DINO-vMF | **66.7** | **39.5** | 37.3 | 12.5 |
| iBOT | 64.1 | 36.6 | 35.2 | 13.4 |
| iBOT-vMF | 65.4 | 38.1 | **38.1** | **13.8** |
| MSN | 56.3 | 29.3 | 31.8 | 11.8 |
| *ViT-Small/16* | | | | |
| DINO | **63.1** | **34.5** | 34.6 | 13.0 |
| DINO-vMF | 62.1 | 33.4 | 34.5 | 12.6 |
| MSN | 61.5 | 33.3 | **36.6** | **15.1** |

Table 7: Video Object Segmentation (VOS)

| Method | $(\mathcal{J}\&\mathcal{F})_m$ | $\mathcal{J}_m$ | $\mathcal{F}_m$ |
| --- | --- | --- | --- |
| *ViT-Base/16* | | | |
| DINO | 62.4 | 60.8 | 64.0 |
| DINO-vMF | **63.4** | 61.6 | **65.2** |
| iBOT | 62.7 | 61.8 | 63.7 |
| iBOT-vMF | 63.1 | **61.9** | 64.2 |
| MSN | 58.0 | 56.1 | 60.0 |
| *ViT-Small/16* | | | |
| DINO | 61.8 | 60.2 | 63.4 |
| DINO-vMF | **62.7** | **60.9** | **64.5** |
| MSN | 59.6 | 57.6 | 61.6 |

versions of the datasets. We observe improved performance with the ViT-Base/16 model when using the vMF versions compared to standard DINO and iBOT on both datasets. However, the ViT-Small/16 model pre-trained with DINO-vMF is only on par or slightly worse than standard DINO pre-training. Further, we find that the MSN pre-trained model for ViT-Small/16 also performs competitively.

**Video Object Segmentation (VOS):** We consider the DAVIS-2017 video instance segmentation benchmark (Pont-Tuset et al., 2017). We use the frozen features from the ImageNet pre-trained model and use a nearest neighbour based segmentation approach following the same experimental protocol as (Caron et al., 2021) and (Jabri et al., 2020). We compare our results with standard DINO, iBOT and MSN in Table 7. We find that the vMF versions show marginal improvements on both ViT backbones. We also find that the MSN transfers poorly to this task compared to DINO and DINO-vMF.

## 6 CONCLUSION

We reinterpreted the DINO (Caron et al., 2021) SSL method as a mixture of von Mises-Fisher distributions in latent space. Based on this interpretation, we proposed to use unnormalized prototypes with appropriate normalization of the component logit scores to enable a more flexible mixture model. Our proposed modifications enable stable training of DINO using larger ViT models and better cluster utilization. The modification requires log-normalizing constants during training. However, using the proposed approximation these are very fast to compute and do not result in any noticeable computational overhead. At inference time the log-normalizer is not used at all. We empirically demonstrated consistent performance gains on a range of transfer tasks. The larger ViT-Base/16 model benefits the most from our proposed modifications and shows significant improvements on all the considered downstream tasks. We accredit this to the fact that, without the proposed modifications, the larger models require unfavourable $L^2$-normalization of the prototypes to obtain stable training. The smaller models are already trained without $L^2$-normalization and for these models we see more marginal gains. Thus, our study has put a spotlight on a seemingly important design choice that, in our opinion, has not been discussed enough in prior work. By improving iBOT using our vMF modification, we show that our work extends to other methods built on the DINO formulation.

DINO-vMF was motivated by a mixture model interpretation of DINO, but we nevertheless use a quite similar training algorithm. For future work it would be interesting to investigate if this interpretation can be used to motivate additional modifications. One could for instance relate the DINO training scheme to stochastic versions of the EM algorithm (Delyon et al., 1999; Celeux & Diebolt, 1992). The soft cluster assignments computed using the teacher network can be viewed as the E-step, where the EMA is similar to the stochastic approximation of the auxiliary quantity used in SAEM (Delyon et al., 1999). The stochasticity comes from mini-batch sampling and multi-crop training. The main difference is in terms of the objective that is optimized in the M-step, where EM maximizes the ELBO whereas DINO(-vMF) minimizes the KL divergence between cluster probabilities. Exploring this connection in more detail could possibly result in better understanding of clustering-based SSL methods, as well as additional performance boosts.

## REPRODUCIBILITY STATEMENT

We intend to make a public release of the code repository and pre-trained models in order to aid the research community to reproduce our experiments. We do not introduce any new hyperparameters in our pre-training method and use the same hyperparameter setup as DINO (Caron et al., 2021) and iBOT (Zhou et al., 2021) which are already publicly available. For other downstream tasks, we closely follow the experimental protocol of other works and explicitly state any differences in the paper.

## ACKNOWLEDGMENTS

This research is financially supported by the Swedish Research Council via the project *Handling Uncertainty in Machine Learning Systems* (contract number: 2020-04122), the Wallenberg AI, Autonomous Systems and Software Program (WASP) funded by the Knut and Alice Wallenberg Foundation, and the Excellence Center at Linköping–Lund in Information Technology (ELLIIT). The computations were enabled by the Berzelius resource provided by the Knut and Alice Wallenberg Foundation at the National Supercomputer Centre.

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

# A  APPENDIX

## A.1  INTUITIVE COMPARISON OF DINO AND DINO-vMF

In Section 3.3, we stated that in a properly normalized formulation, the model cannot naively increase $\|w^{(k)}\|$ to increase the probability of assigning a data sample to a cluster or mixture component. We expand on this idea by providing an intuitive example. Consider a 2-dimensional latent space on the unit circle. Let $y$ denote a vector in this latent space with $\|y\| = 1$. We consider a simple setting with only two clusters, $z \in \{1, 2\}$, and let $w^{(1)}$ and $w^{(2)}$ be prototypes representing these clusters. The DINO formulation proposes the following logit score $l_{\text{DINO}}^{(1)}$ for assigning vector $y$ to cluster 1:

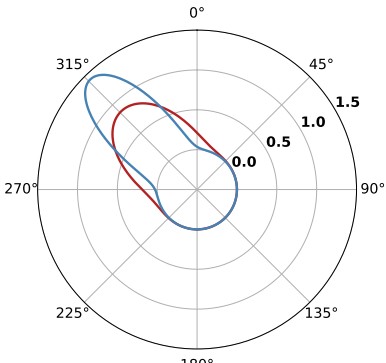

$$l_{\text{DINO}}^{(1)} = \langle w^{(1)}, y \rangle = \|w^{(1)}\| \cos \theta^{(1)}$$

For simplicity, let us ignore the temperature scaling that is applied in the sharpening step. Here, the model can simply increase $\|w^{(1)}\|$ to increase $l_{\text{DINO}}^{(1)}$ as long as $\cos \theta^{(1)} > 0$. Instead, our proposed DINO-vMF uses the following $l_{\text{DINO-vMF}}^{(1)}$ logit scores:

Figure 3: von Mises-Fisher density on the circle for a prototype vector pointing in the direction of $315°$ for two different values of prototype magnitudes (larger magnitude: blue curve, smaller magnitude: red curve).

$$l_{\text{DINO-vMF}}^{(1)} = \langle w^{(1)}, y \rangle + \log C_p(\|w^{(1)}\|) = \|w^{(1)}\| \cos \theta^{(1)} + \log C_p(\|w^{(1)}\|)$$

where $\log C_p(\cdot) < 0$ and $\log C_p(\|w^{(1)}\|)$ monotonically decreases with $\|w^{(1)}\|$. So, merely increasing $\|w^{(1)}\|$ can increase the first term but this is counteracted by the second term. For given values of $\|w^{(1)}\|$ and $\theta^{(1)}$ there exists a threshold $\hat{\theta}^{(1)}$ below which increasing $\|w^{(1)}\|$ results in an increased logit score. The logit score decreases when $\theta^{(1)}$ is larger than this threshold. So, the model needs to decrease $\theta^{(1)}$ by mapping the vector $y$ closer to the prototype in order to benefit from increasing $\|w^{(1)}\|$. We show an illustration of how the vMF density varies with the prototype magnitude in Figure 3. This implies that, if the DINO-vMF model can consistently map a set of images close to a prototype, only then it can benefit from increasing the magnitude of that prototype.

## A.2  VON MISES-FISHER NORMALIZATION CONSTANT

Given a von Mises-Fisher distribution in $p$-dimensions with parameters $\mu$ and $\kappa$, its normalization constant $C_p(\kappa)$ is given by:

$$C_p(\kappa) = \frac{\kappa^{p/2-1}}{(2\pi)^{p/2} I_{p/2-1}(\kappa)}$$

where $I_\nu$ denotes the modified Bessel function of the first kind and order $\nu$. We approximate the $I_{p/2-1}(\kappa)$ term as explained in section 3.3 by using an asymptotic expansion. The softmax-logit scores for each mixture component is obtained by adding the network output and the log-normalization constant corresponding to that component. Because of the softmax formulation, it is sufficient for the log-normalization constant approximation to be correct up to a constant. We compare the values of $\log C_p(\kappa)$ obtained using our approximation to those obtained by using the scipy implementation of $I_\nu$ instead. We consider $\kappa \in [20, 500]$ and in Figure 4 we show a comparison of the values of $\log C_p(\kappa)$ obtained using our approximation. We did produce the same figure with the same values computed using the scipy-based implementation, but the lines appear to completely overlap with our approximation, so this is not shown. We define an approximation error $\epsilon(\kappa)$ with respect to the scipy implementation as follows:

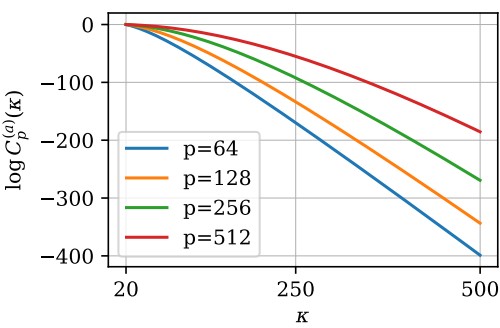

Figure 4: Our approximation up to a constant, $\log C_p^{(a)}(\kappa)$.

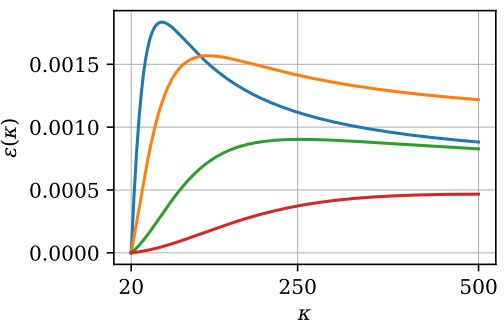

Figure 5: Approximation error in computing $\log C_p(\kappa)$ compared to scipy-based implementation. Note that the error is small compared to the approximated function values shown in Figure 4.

$$\epsilon(\kappa) = \log C_p^{(a)}(\kappa) - \log C_p^{(s)}(\kappa)$$

where $C_p^{(a)}(\kappa)$ denotes our approximation and $C_p^{(s)}(\kappa)$ denotes the scipy-based implementation[5]. The error values are shown in Figure 5 for different bottleneck dimensions $p$ in the DINO prediction head. For $p = 256$, as in the case of DINO, we can observe that the approximation error is small compared to the actual function values.

### A.3 PROTOTYPE UTILIZATION AT DIFFERENT SIMILARITY THRESHOLDS

As explained in section 5.2, we identify unique prototypes based on a cosine similarity threshold. Here, we present a further discussion about how the prototype utilization varies with the choice of similarity threshold. In Figures 6 and 7, we show the number of unique prototypes and the size of the largest duplicate set of prototypes. An unusually large duplicate set indicates a *void prototype set*. We observe that the void prototype set in ViT-base/16 hardly changes in size even when the similarity threshold is increased up to $0.99$. We observe a higher prototype utilization with DINO-vMF compared to DINO for both the ViT-Base and ViT-Small models. Only exception is when we set the similarity threshold to a large value of $0.99$ for ViT-Small/16 model trained with DINO. We expect these near-duplicates to further increase in similarity, if the model training is continued for more epochs.

### A.4 RELATIONSHIP BETWEEN CENTERING AND CLUSTER PRIOR

The centering parameter $c^{(k)}$, computed according to Eq. (8) in DINO or Eq. (9) in DINO-vMF, is related to an estimate of the cluster prior $\pi^{(k)}$ in our mixture model interpretation. Indeed, for DINO, the centering parameter averages the logit scores of the components, scaled by the sharpening parameter $\tau_t$, so $\exp\left[c^{(k)}/\tau_t\right]$ can be viewed as an estimate of $\pi^{(k)}$ up to normalization. For DINO-vMF we similarly average the cluster probability (i.e., responsibility) over a batch at each iteration, and update our estimate of $c^{(k)}$ using an EMA of the corresponding logits; see (9). Note that the sharpening parameter $\tau_t$ is included in the expression for the probabilities, which means that, in this formulation, we can estimate $\pi^{(k)}$ as $\exp\left[c^{(k)}\right]$ up to normalization, i.e. without the scaling $\tau_t$. The key difference between the two estimates is thus whether the averaging is done in logit space or in probability space. The latter is closer to how one would typically estimate $\pi^{(k)}$ in a standard mixture model, e.g. using the EM algorithm, as an average of the responsibility for cluster $k$.

Combining the aforementioned estimates with the expressions for the teacher probabilities (Eq. (4) for DINO or Eq. (7) for DINO-vMF) and comparing with the responsibilities as computed for a mixture model in Eq. (2), it is tempting to simply relate the factors involving $c^{(k)}$ with the prior

---

[5]https://docs.scipy.org/doc/scipy/reference/generated/scipy.special.iv.html

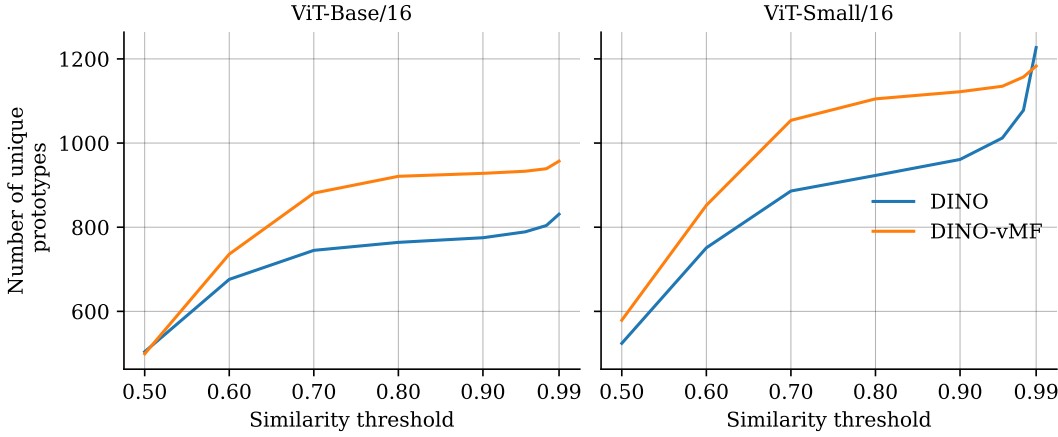

Figure 6: Number of unique prototypes considering different cosine similarity threshold for identifying duplicate prototypes. Even at relatively low similarity thresholds, DINO-vMF utilizes more unique and adequately separated prototypes.

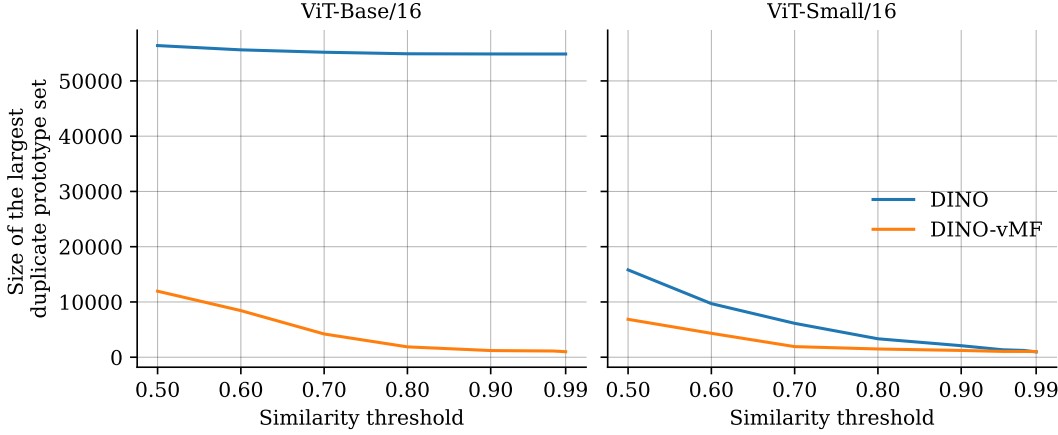

Figure 7: Size of the largest set of duplicate prototypes. DINO pre-trained ViT-Base models contain a large duplicate set, that we call *void prototype set*. This is avoided in DINO-vMF pre-training. Recall that ViT-Base models are trained with $L^2$-normalized prototypes whereas the ViT-Small models are trained with unnormalized prototypes.

probability $\pi^{(k)}$ in (2). However, this would be an incorrect interpretation since there is a minus sign in the exponents in Eqs. (4)and (7).

To understand this discrepancy, note first that in a mixture model we would estimate $\pi^{(k)}$ from the available data, but also encourage data points to be assigned to components with higher $\pi^{(k)}$-values. This can be thought of as a "rich gets richer" approach. If a certain cluster is found to be dominant, then this information is fed back to the estimates of the responsibilities, increasing the probability of assigning data points to this cluster.

Such an approach makes sense when the data is fixed, and we are simply trying to learn a clustering model that fits the data as well as possible. However, it is also distinctly different from the role of the centering in DINO(-vMF), where the data are mapped to latent representations and the clustering is carried out in this latent space. For these models we fix $\pi$ to be uniform. The centering can be viewed as a way to further encourage this uniformity. This is achieved by generating targets using the *inverse* of the estimated cluster probabilities, i.e. instead of multiplying with $\pi^{(k)} \propto \exp\left[c^{(k)}\right]$ we multiply with $1/\pi^{(k)} \propto \exp\left[-c^{(k)}\right]$. This can be viewed as a way to rebalance the component assignments, by encouraging the model to assign more samples to less used components and less samples to more highly used components. We effectively obtain a "rich gets poorer" and "poor gets richer" strategy.

Table 8: ImageNet kNN classification accuracy ablating on the impact of $L^2$-normalization of prototypes, vMF normalization and probability centering for 2 manually seeded trainings.

| | Centering space | | | |
| --- | --- | --- | --- | --- |
| | Run 1 (seed=0) | | Run 2 (seed=42) | |
| Normalization | logit | probability | logit | probability |
| None | 70.43 † | 70.74 | 69.93 † | 70.24 |
| $L^2$ | 69.72 ‡ | 70.18 | 69.30 ‡ | 70.17 |
| vMF | 70.51 | **71.12** | 69.91 | **70.62** |

† = default setting for DINO/ViT-S, ‡ = default setting for DINO/ViT-B.

## A.5 OUR MODIFICATION OF IBOT WITH VMF NORMALIZATION

We incorporate our vMF modification in iBOT Zhou et al. (2021) which uses a DINO-derived loss formulation. The self-supervised pre-training in iBOT uses a self-distillation objective, $\mathcal{L}_{\texttt{[CLS]}}$ and a masked image modeling (MIM) objective, $\mathcal{L}_{[\text{MIM}]}$. Both objectives are formulated as cross-entropy terms similar to DINO based on the teacher and student probability distributions. We apply the vMF modification to the teacher and student probability distributions for the [CLS] and patch tokens as follows:

$$P_s^{\texttt{[CLS]}}(\boldsymbol{x})^{(k)} \overset{k}{\propto} \exp\left[\left\langle \boldsymbol{w}_s^{(k)}, \boldsymbol{y}^{\texttt{[CLS]}} \right\rangle / \tau_s + \log C_p\left(\kappa_s^{(k)}\right)\right],$$

$$P_t^{\texttt{[CLS]}}(\boldsymbol{x})^{(k)} \overset{k}{\propto} \exp\left[-c^{(k)}\right] \exp\left[\left\langle \boldsymbol{w}_t^{(k)}, \boldsymbol{y}^{\texttt{[CLS]}} \right\rangle / \tau_t + \log C_p\left(\kappa_t^{(k)}\right)\right],$$

$$P_s^{\text{patch}}(\boldsymbol{x})^{(k)} \overset{k}{\propto} \exp\left[\left\langle \boldsymbol{w}_s^{(k)}, \boldsymbol{y}^{\text{patch}} \right\rangle / \tau_s + \log C_p\left(\kappa_s^{(k)}\right)\right],$$

$$P_t^{\text{patch}}(\boldsymbol{x})^{(k)} \overset{k}{\propto} \exp\left[-c^{(k)}\right] \exp\left[\left\langle \boldsymbol{w}_t^{(k)}, \boldsymbol{y}^{\text{patch}} \right\rangle / \tau_t + \log C_p\left(\kappa_t^{(k)}\right)\right].$$

Consider image views $\boldsymbol{u}$ and $\boldsymbol{v}$ and $N$ masked views $\hat{\boldsymbol{u}}_i$ and $\hat{\boldsymbol{v}}_i$ with masks $m_i$. Then, the overall loss in iBOT is simply computed as a sum of the objectives, $\mathcal{L}_{\texttt{[CLS]}}$ and $\mathcal{L}_{[\text{MIM}]}$, which are formulated as follows:

$$\mathcal{L}_{\texttt{[CLS]}} = -P_t^{\texttt{[CLS]}}(\boldsymbol{v})^T \log P_s^{\texttt{[CLS]}}(\boldsymbol{u})$$

$$\mathcal{L}_{[\text{MIM}]} = -\sum_{i=1}^{N} m_i \cdot P_t^{\text{patch}}(\boldsymbol{u}_i)^T P_s^{\text{patch}}(\hat{\boldsymbol{u}}_i)$$

## A.6 ADDITIONAL EXPERIMENTS AND DETAILS

### A.6.1 DETAILED ABLATION STUDIES

We run the ablation studies explained in 5.1.1 for 2 runs and all pre-trainings within a run are done with manually fixed seeds to ensure that the results are comparable and reproducible within each run. The results for each run are reported in Table 8

### A.6.2 IMAGENET CLASSIFICATION WITH FULL DATASET

We present additional baseline comparisons for ViT-Small/16 and ViT-Base/16 architectures in Tables 9 and 10 respectively. In addition to the results presented based on DINO and DINO-vMF pre-trained ViT-Small/16 and ViT-Base/16 models, we consider the ViT-Small/8 backbone architecture that uses a smaller patch size of 8. This model is more expensive to train compared to the ViT-Small/16 model.

DINO pre-trained this model for 800 epochs. We conduct a smaller experiment for 40 epochs with the same hyperparameters as in DINO. The results are reported in Table 11. The results are similar to our observations regarding the ViT-Small/16 architecture.

Table 9: ImageNet classification accuracy using linear and kNN classifiers for ViT-Small/16

| Method | kNN | Lin. [C] | Lin. [D] |
|---|---|---|---|
| DINO | 74.44 | 76.09 | 76.98 |
| DINO-vMF | 74.74 | 76.19 | 76.97 |
| MSN | 74.86 | **76.20** | 76.62 |
| SwAV | 66.3 | – | 73.5 |
| iBOT | **75.2** | – | **77.9** |
| MoCo-v3 | – | – | 73.4 |
| MoBY | – | – | 72.8 |
| MoCo-v2 | 64.4 | – | 72.7 |
| BYOL | 66.6 | – | 71.4 |

Table 10: ImageNet classification accuracy using linear and kNN classifiers for ViT-Base/16

| Method | kNN | Lin. [C] | Lin. [D] |
|---|---|---|---|
| iBOT | 77.10 | 79.33 | 79.50 |
| iBOT-vMF | **78.66** | **80.20** | **80.27** |
| DINO | 76.11 | 77.88 | 78.17 |
| DINO-vMF | 77.40 | 78.67 | 78.81 |
| BeIT-v2 | – | – | 80.10 |
| MSN | 73.33 | 75.84 | 74.80 |
| MoCo-v3 | – | – | 76.7 |
| MAE | – | – | 68.0 |
| BeIT | – | – | 56.7 |

Table 11: ImageNet classification accuracy using linear and kNN classifiers for ViT-S/8 architecture

| Method | kNN | Linear (`[CLS]`) | Linear (DINO) |
|---|---|---|---|
| DINO | 69.98 | 72.75 | **74.24** |
| DINO-vMF | **70.41** | **72.91** | 74.14 |

### A.6.3 Implementation details for linear classification on small datasets

We consider the following small datasets: Aircraft (Maji et al., 2013), Caltech101 (Li et al., 2022), CIFAR10, CIFAR100 (Krizhevsky, 2009), DTD (Cimpoi et al., 2014), Flowers (Nilsback & Zisserman, 2008), Food (Bossard et al., 2014), Pets (Parkhi et al., 2012) and SUN397 (Xiao et al., 2010). The linear classifier is trained with $L^2$-regularization and the strength is chosen for each dataset by using 5-fold cross-validation among a set of 45 values spaced linearly in the range $[-6, 5]$ in log-space following Ericsson et al. (2021).

### A.6.4 ImageNet few-shot evaluation

The few-shot evaluation involving 1, 2 and 5 images per class were conducted using three different splits. In Table 12 and Table 13, we show the mean and standard deviation of the top-1 validation classification accuracy over the three splits.

Table 12: ImageNet few-shot evaluation (as in MSN) with ViT-Base/16 model

| | ViT-Base/16 | | | | |
|---|---|---|---|---|---|
| Dataset | DINO | MSN | DINO-vMF | iBOT | iBOT-vMF |
| 1 img/cls | $41.8 \pm 0.3$ | $49.8 \pm 0.2$ | $50.3 \pm 0.2$ | $46.0 \pm 0.3$ | $\mathbf{51.6 \pm 0.1}$ |
| 2 imgs/cls | $51.9 \pm 0.6$ | $58.9 \pm 0.4$ | $59.3 \pm 0.4$ | $56.0 \pm 0.8$ | $\mathbf{61.1 \pm 0.7}$ |
| 5 imgs/cls | $61.4 \pm 0.2$ | $65.5 \pm 0.3$ | $66.1 \pm 0.2$ | $64.7 \pm 0.3$ | $\mathbf{68.3 \pm 0.3}$ |

### A.6.5 Fine-tuning evaluation

We conduct fine-tuning experiments on ImageNet-1K following the same training protocol as Bao et al. (2021) which is found to consistently produce the best fine-tuning results at minimum training epochs Zhou et al. (2021). We train our ViT-S/16 and ViT-B/16 models for 200 and 100 epochs respectively. We report the best result obtained after conducting a sweep over the learning rates:

Table 13: ImageNet few-shot evaluation (as in MSN) with ViT-Small/16 model

| Dataset | ViT-Small/16 | | |
|---|---|---|---|
| | DINO | MSN | DINO-vMF |
| 1 img/cls | $38.9 \pm 0.4$ | $\mathbf{47.1 \pm 0.1}$ | $39.2 \pm 0.6$ |
| 2 imgs/cls | $48.9 \pm 0.3$ | $\mathbf{55.8 \pm 0.6}$ | $49.4 \pm 0.7$ |
| 5 imgs/cls | $58.5 \pm 0.1$ | $\mathbf{62.8 \pm 0.3}$ | $59.1 \pm 0.2$ |

$\{8e\text{-}4, 9e\text{-}4, 1e\text{-}3, 2e\text{-}3\}$. For fine-tuning on smaller datasets, we follow the fine-tuning recipe of Touvron et al. (2021) of training for 1000 epochs with a small learning rate of $7.5e\text{-}6$. We report the accuracy for CIFAR-10, CIFAR-100 and Flowers datasets and for the Aircraft dataset, we report the mean-per-class accuracy in Tables 14 and 15. We observe that the vMF variants are mostly on-par or slightly better than their standard counterparts. While performance on CIFAR-10, CIFAR-100 and Flowers are beginning to saturate, we observe greater performance gains on the Aircraft dataset.

Table 14: Fine-tuning results with ViT-Small/16 after pre-training on ImageNet-1K

| Method | CIFAR10 | CIFAR100 | Flowers | Aircraft | ImageNet-1K |
|---|---|---|---|---|---|
| Random (DeIT) | 99.0 | 89.5 | 98.2 | – | 79.9 |
| Random (DeIT-III) | – | – | – | – | 81.4 |
| BeIT | 98.6 | 87.4 | 96.4 | – | – |
| DINO | 99.0 | 90.5 | 98.5 | 84.2 | 82.0 |
| DINO-vMF | 99.0 | **90.7** | **98.7** | 85.0 | 81.8 |
| iBOT | **99.1** | **90.7** | 98.6 | **85.7** | **82.3** |

Table 15: Fine-tuning results with ViT-Base/16 after pre-training on ImageNet-1K

| Method | CIFAR10 | CIFAR100 | Flowers | Aircraft | ImageNet-1K |
|---|---|---|---|---|---|
| Random (DeIT) | 99.0 | 90.8 | 98.4 | – | 81.8 |
| Random (DeIT-III) | – | – | – | – | 83.8 |
| BeIT | 99.0 | 90.1 | 98.0 | – | 83.4 |
| DINO | 99.1 | 91.7 | 98.8 | 84.5 | 83.6 |
| DINO-vMF | **99.2** | 91.9 | **98.9** | 84.8 | 83.6 |
| iBOT | **99.2** | 92.2 | **98.9** | 85.1 | 84.0 |
| iBOT-vMF | **99.2** | **92.6** | 98.8 | **85.8** | **84.1** |
| MAE | – | – | – | – | 83.6 |

