# OpenReview forum: "DINO as a von Mises-Fisher mixture model"
_ICLR.cc/2023/Conference — ICLR 2023 notable top 25%_

### Official Review · Reviewer_njCU · 2022-10-25

**Confidence:** 3
**Correctness:** 3
**Technical Novelty And Significance:** 3
**Empirical Novelty And Significance:** 3
**Recommendation:** 8

**Clarity, Quality, Novelty And Reproducibility:**

The paper is well written and authors claim they will release code and pre-trained models.

Minor details:

Section 3.1
- Column notation is not consistent with vector notation in general. Column should be denoted as, for example, $\mathbf{w}^{(k)}$ instead of $W^{(k)}$. In general, notation for vectors and matrices should be improved.

Section 3.2
- $|\mu| = 1$ should be replaced with $\|\mu \|_2$

**Strength And Weaknesses:**

The paper proposes a very interesting interpretation of DINO and the proposed modification increases flexibility of the method by including per-cluster shape parameter. The experimental section includes a number of results to demonstrate the effectiveness of the proposed method.

**Summary Of The Paper:**

The paper provides an interpretation of DINO as a mixture model with components with a circular von Mises-Fisher distribution.
Using this interpretation, the authors propose a normalization when computing cluster assignments, which improves stability and flexibility of the mixture model.

**Summary Of The Review:**

A useful and well-written paper.

---

> ### Author Response · Authors · 2022-11-17
> **Response to reviewer njCU**
>
> We thank the reviewer for the useful feedback and we appreciate that the reviewer found the interpretation of DINO to be interesting. We address the few concerns raised by the reviewer below:
>
> > Q1. Column notation is not consistent with vector notation in general. In general, notation for vectors and matrices should be improved.
>
> We have made minor updates to the notations for matrices, vectors and column vectors. We believe that the new notation choice is more clear to follow.
>
> > Q2. |\mu|=1 should be replaced with |\mu|_2
>
> Since we use the 2-norm frequently in the paper we have decided not to include the subscript “2” throughout to avoid a cluttered notation. However, we have changed from single | to double || to emphasize that we are dealing with a norm. We believe that this modification should be sufficient to avoid any confusion.

---

### Official Review · Reviewer_aACz · 2022-10-25

**Confidence:** 4
**Correctness:** 4
**Technical Novelty And Significance:** 2
**Empirical Novelty And Significance:** 2
**Recommendation:** 6

**Clarity, Quality, Novelty And Reproducibility:**

- Clarity :: This paper is easy to follow.
- Quality :: Empricial results are not convincing.
- Novelty :: I think the proposed method is novel.
- Reproducibility :: Implementation details are provided.


**Strength And Weaknesses:**

Strengths
- The interpretation is quite interesting, and the proposed modification makes sense.
- This paper is easy to follow and well-written.

Weaknesses
- My major concern is about empirical results.
  - This paper should compare the proposed method, DINO-vMF, with other SSL methods. The comparison with only DINO and MSN is not enough, and it makes some difficulty to check whether the obtained performance is good or not.
  - The obtained gains seem marginal. For example, only 0.7\% accuracy gain on ImageNet using ViT-Base, and no gain when using ViT-Small. Also, there is no meaningful gain in transfer learning (e.g., Table 5).
  - There is no fine-tuning experiment.
- Another concern is that this paper is too limited to DINO. This limited applicability would be critical in this field because there have been developed many SSL methods after DINO, e.g., MAE and iBOT. It would be better if this paper shows that the interpretation can be incorporated into other SSL frameworks.


**Summary Of The Paper:**

This paper suggests interpreting the recent self-supervised learning framework, DINO, as a mixture model of von Mises-Fisher components. Based on the interpretation, this paper proposes a modification to encourage DINO training to be more flexible and stable. This paper demonstrates the effectiveness of the modification under various experiments.


**Summary Of The Review:**

Although the interpretation and the proposed modification is convincing, the empirical results are too marignal and not interesting. Hence I vote for weak rejection.

---

> ### Author Response · Authors · 2022-11-17
> **Response to reviewer aACz**
>
> We thank the reviewer for the insightful and valuable feedback that was useful to improve our paper. We appreciate that the reviewer found our interpretation of DINO [1] to be interesting. We address the concerns raised by the reviewer below:
>
> > Q1. This paper should compare the proposed method, DINO-vMF, with other SSL methods. The comparison with only DINO and MSN is not enough, and it makes some difficulty to check whether the obtained performance is good or not.
>
> We have included a comparison with iBOT [3] as well, which is a recent state-of-the-art method. See the reply below (in Q4) for further details.
>
> > Q2. The obtained gains seem marginal. For example, only 0.7% accuracy gain on ImageNet using ViT-Base, and no gain when using ViT-Small. Also, there is no meaningful gain in transfer learning (e.g., Table 5).
>
> We agree that the gains are marginal when using ViT-Small. We accredit this to the fact that ViT-Small is trained using prototypes that are not $L^2$ normalized. From our ablation studies, it can be seen that the maximal improvement in performance is obtained when we compare the “vMF normalization + centering in probability space” to the default setting for ViT-Base (“$L^2$ normalization + centering in logit space”). For ViT-Small, the default setting uses “no normalization + centering in logit space” in DINO as well as its extensions. When comparing these settings, we observe a smaller performance improvement in the ablation studies. However, simply skipping the normalization is not a generally applicable solution since it can lead to training instabilities for larger models. The vMF normalization is a way to address this limitation and remove the need for adapting the normalization to the choice of encoder architecture (ViT-Small/Base). DINO does not consider models larger than ViT-Base but iBOT reports results for ViT-Large and this uses the same setting as ViT-Base. We expect our vMF interpretation to improve this result but we do not have the resources to run this experiment.
>
> > Q3. There is no fine-tuning experiment.
>
> In this paper, we focus on evaluating the quality of the learned representations through kNN, linear probing and few-shot learning experiments. We agree that a fine-tuning experiment would have been interesting as well, but considering the space (and time) limitations we have not prioritized this, since we believe that the aforementioned set of experiments are sufficient to illustrate the effectiveness of the proposed method.
>
> > Q4. Another concern is that this paper is too limited to DINO. This limited applicability would be critical in this field because there have been developed many SSL methods after DINO, e.g., MAE and iBOT. It would be better if this paper shows that the interpretation can be incorporated into other SSL frameworks.
>
> We incorporated vMF normalization in the iBOT formulation and trained a ViT-Base/16 model with the same hyperparameter setup as iBOT. Through this experiment, we show that our reinterpretation of DINO as a vMF mixture model in the latent space is more generally applicable and that it can be used with other clustering-based SSL methods as well. Furthermore, the new results show that the proposed modification indeed results in a significant improvement of the performance of iBOT on the various downstream tasks (kNN top-1 validation accuracy improved from 77.1% with iBOT to 78.66% with iBOT-vMF, linear classification accuracy improved from 79.5% with iBOT to 80.27% with iBOT-vMF). We have updated the experiments section with results for both standard iBOT and iBOT-vMF. DINO is an influential self-supervised learning work that has inspired other state-of-the-art methods such as EsViT [2], MSN [4] and iBOT that use a similar formulation involving $L^2$ normalized prototypes for larger ViT models. For this reason, our interpretation is relevant to other DINO-derived SSL methods and can potentially improve other methods as well. We have emphasized this in the new version of our paper.
>
> **References**
>
> [1] Mathilde Caron, Hugo Touvron, Ishan Misra, Hervé Jégou, Julien Mairal, Piotr Bojanowski, and Armand Joulin. Emerging properties in self-supervised vision transformers. In ICCV, 2021.
>
> [2] Chunyuan Li, Jianwei Yang, Pengchuan Zhang, Mei Gao, Bin Xiao, Xiyang Dai, Lu Yuan, and Jianfeng Gao. Efficient self-supervised vision transformers for representation learning. In ICLR, 2021.
>
> [3] Jinghao Zhou, Chen Wei, Huiyu Wang, Wei Shen, Cihang Xie, Alan Yuille, and Tao Kong. Image bert pre-training with online tokenizer. In ICLR, 2021.
>
> [4] Mahmoud Assran, Mathilde Caron, Ishan Misra, Piotr Bojanowski, Florian Bordes, Pascal Vincent, Armand Joulin, Michael Rabbat, and Nicolas Ballas. Masked siamese networks for label-efficient learning. arXiv preprint arXiv:2204.07141, 2022.

---

> > ### Comment · Reviewer_aACz · 2022-12-01
> > **Response to Authors**
> >
> > Thank you for your time and efforts in your response.
> >
> > First of all, thank you for the additional experiments with iBOT. I have checked that the proposed technique can be applied to another framework, iBOT. However, iBOT also directly uses the DINO objective, so my concern about the heavy reliance on DINO is still remaining. Furthermore, I'm still wondering why the authors do not include other SSL methods into the main table. The current baselines are still limited and I strongly suggest adding more baselines (with both ResNet and ViT backbones) into the main table (e.g., Table 1 in the iBOT paper). Also, I think conducting fine-tuning experiments is important in the self-supervised learning literature. (Also, IMO, this paper has enough space to add more tables.)
> >
> > Hence, I would like to keep my rating because some of my concerns are still remaining.

---

> > > ### Author Response · Authors · 2022-12-08
> > > **Further response to reviewer aACz (part 1/3)**
> > >
> > > Thank you for reviewing our previous response. With this response, we provide a few clarifications and additional results to address the remaining concerns.
> > >
> > > ### Regarding our heavy reliance on DINO:
> > >
> > > It is true that our re-interpretation and improvements are developed specifically with DINO in mind, and they are therefore only applicable to DINO and methods that are similar to DINO. There seems to be some misunderstanding that our modification should be applicable to other types of SSL methods as well, but this is not the case and we never claim this in our paper. However, considering that the DINO formulation is used in several state-of-the-art methods we still believe that improvements to the __DINO framework__ are highly relevant. iBOT is one of the two methods that you specifically brought up in your original review, and based on this useful feedback we have also illustrated the benefit of our method in the context of iBOT, which we believe has significantly improved the paper – thanks for this suggestion! Developing similar probabilistic re-interpretations of other SSL frameworks is an interesting direction for future work, but beyond the scope of our paper.
> > >
> > > ### Adding more baselines to our main table:
> > >
> > > We can certainly add more SSL methods to the main table. We did not do this mainly because of page limit constraints. Since we propose a method to improve DINO and other methods based on the DINO framework, we focused on these standard methods as our primary baselines. We provide a more extensive comparison in the tables below, categorized based on the model architecture. We will fit in a selection of these baselines in Table 2 in the camera-ready version and add the remaining baselines to the appendix.
> > >
> > > __ViT-Small/16 (21M parameters, throughput: 1007 im/s)__
> > >
> > > | Method                     | kNN accuracy (%) | Linear accuracy (%) |
> > > |----------------------------|------------------|---------------------|
> > > | BYOL [Grill et al. 2020]   | 66.6             | 71.4                |
> > > | MoCo-v2 [Chen et al. 2020] | 64.4             | 72.7                |
> > > | MoBY [Xie et al. 2021]     | --               | 72.8                |
> > > | MoCo-v3 [Chen et al. 2021] | --               | 73.4                |
> > > | SwAV [Caron et al. 2020]   | 66.3             | 73.5                |
> > > | MSN [Assran et al. 2022]   | 74.9             | 76.6                |
> > > | DINO [Caron et al. 2021]   | 74.5             | 77.0                |
> > > | DINO-vMF [Ours]            | 74.7             | 77.0                |
> > > | iBOT [Zhou et al. 2021]    | 75.2             | 77.9                |
> > > ***
> > >
> > > __ViT-Base/16 (85M parameters, throughput: 312 im/s)__
> > >
> > > | Method                     | kNN accuracy (%) | Linear accuracy (%) |
> > > |----------------------------|------------------|---------------------|
> > > | BeIT [Bao et al. 2021]     | --               | 56.7                |
> > > | MAE [He et al. 2022]       | --               | 68.0                |
> > > | MSN [Assran et al. 2022]   | 73.3             | 74.8                |
> > > | MoCo-v3 [Chen et al. 2021] | --               | 76.7                |
> > > | DINO [Caron et al. 2021]   | 76.1             | 78.2                |
> > > | DINO-vMF [Ours]            | 77.4             | 78.8                |
> > > | iBOT [Zhou et al. 2021]    | 77.1             | 79.5                |
> > > | BeIT-v2 [Peng et al. 2022] | --               | 80.1                |
> > > | iBOT-vMF [Ours]            | 78.7             | 80.3                |
> > > ***
> > >
> > > __ViT-Large/16 (307M parameters, throughput: 102 im/s)__
> > >
> > > | Method                  | kNN accuracy (%) | Linear accuracy (%) |
> > > |-------------------------|------------------|---------------------|
> > > | MAE [He et al. 2022]    | --               | 73.5                |
> > > | BeIT [Bao et al. 2021]  | --               | 73.5                |
> > > | iBOT [Zhou et al. 2021] | 78.0             | 81.0                |
> > > ***

---

> > > ### Author Response · Authors · 2022-12-08
> > > **Further response to reviewer aACz (part 2/3)**
> > >
> > > Continuing from our previous comment providing baselines for ViTs, we provide baselines for the ResNet50 backbone in the table below to make it easier to compare with ViT baselines.
> > >
> > > __ResNet50 (23M parameters, throughput: 1237 im/s)__
> > >
> > > | Method                             | kNN accuracy (%) | Linear accuracy (%) |
> > > |------------------------------------|------------------|---------------------|
> > > | BarlowTwins [Zbontar et al. 2021]  | 66.0             | 73.2                |
> > > | BYOL [Grill et al. 2020]           | 64.8             | 74.4                |
> > > | MoCo-v3 [Chen et al. 2021]         | --               | 74.6                |
> > > | SimCLR-v2 [Chen et al. 2020]       | --               | 74.6                |
> > > | DeepCluster-v2 [Caron et al. 2020] | 67.1             | 75.2                |
> > > | SwAV [Caron et al. 2020]           | 65.7             | 75.3                |
> > > | DINO [Caron et al. 2021]           | 67.5             | 75.3                |
> > > | SMoG [Pang et al. 2022]            | --               | 76.4                |
> > > | CoKe [Qian et al. 2022]            | --               | 76.4                |
> > >
> > > ### Regarding other backbone architectures
> > >
> > > Based on the suggestion of reviewer SyZr to add results with other backbone models, including other ViT variants, we had already carried out additional experiments using the ViT-Small/8 (refer section A.5.2 and Table 9 in the appendix). These new results are in line with our observations regarding the ViT-Small/16 architecture (both are trained without L2 normalization in DINO). Our explanation to 2nd question (Q2) in our first response (on 17 Nov 2022) is supported by these new findings.
> > >
> > > DINO can be trained with a ResNet backbone, but other methods based on DINO such as iBOT and EsViT do not directly support a ResNet backbone. Their formulation could be adjusted to support a ResNet backbone but this has not been studied. A ResNet50 backbone pre-trained using DINO achieved much worse performance than the comparable ViT-Small/16 architecture (kNN accuracy: 67.5% vs 74.5%, linear probing accuracy: 75.3% vs 77.0%). As Vision Transformers have shown to be more promising in this line of work, we only explore different variations of Vision Transformers. We have not considered the ViT-Large/16 backbone as we do not have the resources to run this experiment.

---

> > > ### Author Response · Authors · 2022-12-08
> > > **Further response to reviewer aACz (part 3/3)**
> > >
> > > ### Adding finetuning experiments:
> > >
> > > Linear probing, kNN and transfer learning experiments focus on evaluating the quality of the learned image representations. Finetuning experiments initialize the network with the pre-trained network weights and finetune the entire network weights based on a finetuning recipe and hyperparameter configuration. Table 11 in iBOT shows that the finetuning results can vary based on the chosen finetuning recipe. On a preliminary experiment on CIFAR100, we found it difficult to reproduce the DINO and iBOT finetuning results unless we used the exact same global batch size of 768 (96 x 8 GPUs). In the past two weeks, we have made a fair attempt at including the finetuning experiments on ImageNet, CIFAR-10, CIFAR-100, Flowers and Aircraft datasets following the finetuning recipe of iBOT which seems to produce the best results compared to other finetuning recipes in their experiments. The results are shown in the tables below.
> > >
> > > Finally, we would like to emphasize that the main motivation for our work was initially not to come up with a new method for improving benchmark performance by a percentage or some fractions of a percentage. Instead, we wanted to (1) provide a better understanding of DINO and DINO-like methods based on a new interpretation, and (2) make the training more consistently stable and less reliant on the seemingly ad hoc design choice of normalizing or not normalizing the prototypes. The improved benchmark performance is a nice side effect of this development.
> > >
> > > __Finetuning evaluation: ViT-Small/16__
> > >
> > > | Method                                  | CIFAR-10 | CIFAR-100 | Flowers  | Aircraft | ImageNet-1K |
> > > |-----------------------------------------|----------|-----------|----------|----------|-------------|
> > > | Random (DeIT) [Touvron et al. 2021]     | 99.0     | 89.5      | 98.2     | --       | 79.9        |
> > > | Random (DeIT-III) [Touvron et al. 2022] | --       | --        | --       | --       | 81.4        |
> > > | BeIT [Bao et al. 2021]                  | 98.6     | 87.4      | 96.4     | --       | --          |
> > > | DINO [Caron et al. 2021]                | 99.0     | 90.5      | 98.5     | 84.2     | 82.0        |
> > > | DINO-vMF [Ours]                         | 99.0     | __90.7__  | __98.7__ | 85.0     | 81.8        |
> > > | iBOT [Zhou et al. 2021]                 | __99.1__ | __90.7__  | 98.6     | __85.7__ | __82.3__    |
> > >
> > > __Finetuning evaluation: ViT-Base/16__
> > >
> > > | Method                                  | CIFAR-10 | CIFAR-100 | Flowers  | Aircraft | ImageNet-1K |
> > > |-----------------------------------------|----------|-----------|----------|----------|-------------|
> > > | Random (DeIT) [Touvron et al. 2021]     | 99.0     | 90.8      | 98.4     | --       | 81.8        |
> > > | Random (DeIT-III) [Touvron et al. 2022] | --       | --        | --       | --       | 83.8        |
> > > | BeIT [Bao et al. 2021]                  | 99.0     | 90.1      | 98.0     | --       | 83.4        |
> > > | DINO [Caron et al. 2021]                | 99.1     | 91.7      | 98.8     | 84.5     | 83.6        |
> > > | DINO-vMF [Ours]                         | __99.2__ | 91.9      | __98.9__ | 84.8     | 83.6        |
> > > | iBOT [Zhou et al. 2021]                 | __99.2__ | 92.2      | __98.9__ | 85.1     | 84.0        |
> > > | iBOT-vMF [Ours]                         | __99.2__ | __92.6__  | 98.8     | __85.8__ | __84.1__    |
> > > | MAE [He et al. 2022]                    | --       | --        | --       | --       | 83.6        |
> > > | SimMIM [Xie et al. 2022]                | --       | --        | --       | --       | 83.8        |
> > >
> > > The vMF variants are mostly on-par or slightly better than the standard counterparts. Note that iBOT achieved a significant performance improvement compared to DINO (~1% improvement on kNN and linear probing accuracies). Still, the difference between their finetuning accuracies are marginal. Also, the finetuning performance on datasets such as CIFAR-10 and Flowers are beginning to saturate. Hence, we include an additional evaluation on the Aircrafts dataset where we observed a greater scope to improve from the linear probing results. The vMF modification does produce significant improvement on the Aircraft dataset compared to the non-vMF counterparts. Marginal improvements are observed for few other datasets. We find these results to be reasonable considering our observed improvements on linear probing and kNN evaluations.

---

> > > > ### Comment · Reviewer_aACz · 2022-12-08
> > > > **Response to Authors**
> > > >
> > > > Thank you for your efforts in this response, especially about the additional fine-tuning experiments, comparisons with other SSL methods, and further clarification/explanation of the contribution. Although I'm still concerned about the marginal improvements, I agree with your points: (1) this work can provide a better understanding of DINO, and (2) can improve training stability. Hence, I have increased my rating.

---

### Official Review · Reviewer_SyZr · 2022-10-25

**Confidence:** 3
**Correctness:** 3
**Technical Novelty And Significance:** 3
**Empirical Novelty And Significance:** 3
**Recommendation:** 8

**Clarity, Quality, Novelty And Reproducibility:**

Overall, the paper is clear and easy to understand (with the exception of Section 5.2.1, as stated above). I would also suggest the authors to put Equations 8 and 9 in a more prominent position, such right before Section 3.4 (so at the point where they first mention how they alter the logits).

As stated above, I believe the insight provided by this paper regarding the vMF formulation of DINO is original. However, I also believe that the algorithmic novelty derived by this insight in this paper is somewhat limited, since it consists of only a small alteration in the training procedure.

Regarding reproducibility, the authors use hyperparameters already publicly available for the training of their models.

**Strength And Weaknesses:**

Strengths:

- The interpretation of DINO as a mixture model is interesting. The authors formalize an interpretation of DINO as a mixture model based on the probabilities that the student and the teacher assign to each cluster, for each given sample. This formalization provides a nice theoretical interpretation of DINO, leading to an alternative design choice for normalization, namely normalizing the distribution of the mixture model arising from DINO. This is more theoretically founded than simple $L_2$ normalization of the representations.

- The authors perform an extensive set of experiments to examine whether their approach improves over the baseline DINO formulation. More specifically, they evaluate the performance of their approach on the downstream ImageNet classification task, the few-shot learning capabilities of their method, as well as the transferability of their learned representations on a given task. This suite of experiments provides a sufficient understanding of the capabilities of their proposed method, and can be used to compare it to regular DINO, as well as other baselines.

Weaknesses:
- The main issue I have with this work is that the novelty of the proposed method is limited, in that it consists of simply reparametrizing DINO by changing the normalization used. I believe that as it stands, the work focuses a lot on this change in normalization, without enough evidence on why it is important aside from the provided experiments. If possible, I believe that incorporating a simple theoretical example on why regular DINO may fail while the proposed DINO-vMF may succeed would greatly improve the paper and further justify the proposed method.

- Regarding the experiments, the benefit over regular DINO is not clear, and seems to be very dependent on the architecture, as can be seen in the Tables in the paper (where DINO-vMF underperforms in the smaller ViT-Small model). Namely, the benefits seem to only be apparent for large models, as stated by the authors. Since there seems to be a major effect derived by the architecture of the backbone of DINO, I believe that the authors should also examine different architectures for this backbone (different ViT sizes, or ResNets).

- As a more minor issue, the section in which the authors examine the void prototypes is somewhat unclear in my opinion. The way I understand it, the authors state that DINO has a tendency to assign a large number of samples to similar representation, while DINO-vMF tends to spread the samples further apart. I would be grateful if the authors could clarify whether this is correct.

**Summary Of The Paper:**

This paper demonstrates an interpretation of DINO as a von Mises-Fisher mixture model on the unit hypersphere. Using this interpretation, the authors alter the architecture of DINO in order to incorporate a normalization term directly in the distribution, and keep the embeddings unnormalized. Using this approach, the authors demonstrate an improvement over the performance of the version of DINO that uses normalized embeddings.

**Summary Of The Review:**

Overall, I feel that this paper provides an interesting interpretation of DINO that leads to an interesting trick to improve its performance. I lean slightly towards acceptance due to the clean addition of this method to regular DINO training, but I believe that the paper can overall be improved due its somewhat limited novelty.

---

> ### Author Response · Authors · 2022-11-17
> **Response to reviewer SyZr (part 1/2)**
>
> We thank the reviewer for the insightful and detailed feedback. We appreciate that the reviewer found our interpretation to be interesting and for considering our experiments to be extensive. We address the concerns raised by the reviewer below:
>
> > Q1. The main issue I have with this work is that the novelty of the proposed method is limited, in that it consists of simply reparametrizing DINO by changing the normalization used. I believe that as it stands, the work focuses a lot on this change in normalization, without enough evidence on why it is important aside from the provided experiments. If possible, I believe that incorporating a simple theoretical example on why regular DINO may fail while the proposed DINO-vMF may succeed would greatly improve the paper and further justify the proposed method.
>
> We provide a brief explanation about how DINO-vMF operates differently from standard DINO in the first paragraph in section 3.3. We have now added a section in the appendix to expand this idea further with an illustrative toy example involving only 2 clusters/mixture components. Of course, this is not a rigorous theoretical proof for why the vMF normalization is favorable (which would likely be difficult to obtain considering that there is little theoretical support for DINO in the first place), but we believe that it gives an intuitive explanation for the proposed modification. Furthermore, as explained in the general comment above, we now include experimental results also for the iBOT method where vMF normalization gives similar performance boosts as for DINO, providing additional empirical support for the modification.
>
> > Q2. Regarding the experiments, the benefit over regular DINO is not clear, and seems to be very dependent on the architecture, as can be seen in the Tables in the paper (where DINO-vMF underperforms in the smaller ViT-Small model). Namely, the benefits seem to only be apparent for large models, as stated by the authors. Since there seems to be a major effect derived by the architecture of the backbone of DINO, I believe that the authors should also examine different architectures for this backbone (different ViT sizes, or ResNets).
>
> First, we would like to emphasize that there is a key difference between DINO-ViT/Small and DINO-ViT/Base, since the former (by default, see https://github.com/facebookresearch/dino#boosting-dino-performance-t-rex) does not $L^2$-normalize the prototypes whereas the latter does. As we show in the ablation study in Table 1, removing the $L^2$ normalization can result in a performance boost, which explains why the difference between DINO and DINO-vMF is not so large when using the ViT/Small model. (Going from no normalization to vMF normalization gives an additional performance boost according to Table 1, though.) However, simply skipping the normalization is not a generally applicable solution since it can lead to training instabilities. The vMF normalization is a way to address this limitation and remove the need for adapting the normalization to the choice of encoder architecture.
>
> Second, as per the suggestion, we have added an experiment involving another ViT architecture, namely the ViT-Small/8 model. The results are in agreement with the ones obtained with ViT-Small/16.

---

> > ### Comment · Reviewer_SyZr · 2022-11-19
> > **Thank you for your responses.**
> >
> > I am grateful to the authors for responding to my comments. I believe that the inclusion of experiments on different ViT sizes and the fact that a similar formulation on a different technique (IBOT) shows similar benefits demonstrates that the vMF formulation is more generally applicable than I previously thought. For completeness, I would encourage the authors to include the equations corresponding to (3), (4), (6), (7) for iBot in the Appendix (even if they have similar form to the ones for DINO, due to how related the techniques are).
> >
> > Furthermore, I am grateful for the additional explanation the authors provided with respect to the void prototypes, and I now fully understand the point they make. I believe that incorporating parts of the above response in the relevant paragraph of Section 5.2 will make it clearer, and I encourage the authors to do so.
> >
> > Overall, as my concerns have been adressed, I have raised my score.

---

> > > ### Author Response · Authors · 2022-12-08
> > > **Thanks for the additional suggestions**
> > >
> > > We thank the reviewer for additional useful suggestions to improve the content of our paper. We agree that both the suggestions can improve the clarity of our paper. It is correct that the iBOT formulation of the loss terms are similar to that of DINO. We will still add a section in the appendix to provide the equations for iBOT-vMF corresponding to the equations (3), (4), (6), (7) in our paper. We will incorporate parts of our answer to Q3 above in Section 5.2, to the extent possible within the page limit constraints.

---

> ### Author Response · Authors · 2022-11-17
> **Response to reviewer SyZr (part 2/2)**
>
> > Q3. As a more minor issue, the section in which the authors examine the void prototypes is somewhat unclear in my opinion. The way I understand it, the authors state that DINO has a tendency to assign a large number of samples to similar representation, while DINO-vMF tends to spread the samples further apart. I would be grateful if the authors could clarify whether this is correct.
>
> Sorry, this interpretation is incorrect. Each mixture component is related to a prototype vector. The prototype vector is used to compute the logit scores for assigning a data sample to a mixture component as shown in Equations 3 and 4 for DINO and Equations 6 and 7 for DINO-vMF (numbering as in the newer paper version). We consider a prototype to be a void prototype if no data samples are “assigned” to the mixture component corresponding to that prototype, where in this analysis a datapoint is considered to be “assigned” to the prototype with the largest probability (i.e. responsibility). What this investigation shows is that DINO
> based on the ViT-Base model tends to produce a large number of void clusters, and furthermore that these void prototypes all converge to the same point (this is what we mean by a prototype set). Specifically, the void prototypes converge to a vector pointing in the opposite direction from mean of latent representation vectors averaged over the training data. That is, since the model does not assign any data points to these prototypes, it tries to move them as far away as possible from the data representations to minimize their effect on the loss. We hypothesize that this results in an unfavorable “waste” of the available prototypes and empirically we have found that DINO-vMF mitigates this issue.

---

### Official Review · Reviewer_Xj5D · 2022-10-25

**Confidence:** 3
**Correctness:** 4
**Technical Novelty And Significance:** 3
**Empirical Novelty And Significance:** 3
**Recommendation:** 8

**Clarity, Quality, Novelty And Reproducibility:**

The paper is well motivated and written, easy to follow the main logic. The vMF interpretation applied to vision is not entirely novel, such as also cited by the authors to Hasnat et al, yet applying to self-distillation is to my opinion new and helpful. The code is not provided, however, and thus is hard to evaluate reproducibility, given the implementation of probabilistic models is by no means without certain complication.


**Strength And Weaknesses:**

Strength: The interpretation of vMF is appealing, especially leading to removing the L2 regularization to enable more flexibility. Apart from the representative strength, the results are also strong, even compared to the recent works such as MSN.

Weaknesses:

1/ An important question is how practical this solution is: many well-understood and interpretable frameworks seem to be less competitive to their counterpart deep models because of their speed. It seems even with necessary approximations such as in 3.3, it is not trivial for the case of other formulas, which might lead to further approximation to, e.g., trade for more speed.
Likewise, the accuracy in the tasks given are appealing, yet how about the other performance aspects such as training time, inference time and else? In my humble opinion, even some discussions on the drawback of this solution would make the paper more–not less–convincing.

2/ Since the normalization of prototypes is key to this solution, various related studies have been conducted esp. in Section 5. However, I suggest Table 1 should be extended to be more convincing in 2 aspects. First, in this small scale with this large dataset, running some rounds and get the variances as well would be more helpful. Second, do it with small datasets or any others, just to confirm the consistent practical reflection of the intuition and explanations given.


**Summary Of The Paper:**

The paper extends the previous work DINO by interpreting it a new way, as a von Mises-Fisher mixture model, and in turn proposes DINO-vMF model and achieves very good performance on numerous tasks.


**Summary Of The Review:**


The work has an intuitive and well-motivated approach based on DINO, which shows impressive results on various tasks. The authors also conducted meaningful studies and analysis to back their intuition and rationale of their proposed method.

---

> ### Author Response · Authors · 2022-11-17
> **Response to reviewer Xj5D**
>
> We thank the reviewer for the insightful feedback and valuable suggestions. We appreciate that the reviewer found our interpretation to be appealing and for commenting on the strong results. We address the concerns raised by the reviewer below:
>
> > Q1. An important question is how practical this solution is: many well-understood and interpretable frameworks seem to be less competitive to their counterpart deep models because of their speed. It seems even with necessary approximations such as in 3.3, it is not trivial for the case of other formulas, which might lead to further approximation to, e.g., trade for more speed. Likewise, the accuracy in the tasks given are appealing, yet how about the other performance aspects such as training time, inference time and else? In my humble opinion, even some discussions on the drawback of this solution would make the paper more–not less–convincing.
>
> Our vMF interpretation is general to methods derived from DINO. The method involves computing a log-normalization constant term as shown in section 3.3. This only depends on the prototype magnitudes and does not depend on the batch size or data volume. The approximation of the log-normalizer that we propose is very fast to compute. The centering operation already exists in DINO and DINO-derived methods and our proposal for centering is not drastically different in terms of computational cost. Hence, our proposed modifications will not add any noticeable overheads compared to the method that is being modified and the training times remain unchanged. At inference time, only the backbone ViT model is used to encode images to vector representations and this is independent of our proposed changes. So, the inference time is exactly the same as the ViT model inference time. We have added a sentence in the conclusions section to clarify that the training and inference time is not affected by the proposed modification.
>
> We agree that it is useful to discuss the limitations of a paper.  However,  we do not discuss the limitations in detail in the paper owing to page constraints. We discuss one limitation of our work in the conclusion section. Though we provide this mixture model interpretation of DINO, we still use the DINO training algorithm with some modifications. One could take inspiration from classical methods to learn mixture models to develop a method that is better adapted for this task.
>
> > Q2. Since the normalization of prototypes is key to this solution, various related studies have been conducted esp. in Section 5. However, I suggest Table 1 should be extended to be more convincing in 2 aspects. First, in this small scale with this large dataset, running some rounds and get the variances as well would be more helpful. Second, do it with small datasets or any others, just to confirm the consistent practical reflection of the intuition and explanations given.
>
> We conducted an extra run of the ablation experiments. We have updated the table with the average performance metrics obtained from the 2 runs. (We understand that 2 runs is not enough to get statistical significance in the results, but it gives some indication. Even for the small ablation study the training time is not insignificant so this is all that we had time for.) In the appendix, we include the results from both runs. All experiments within a run use the same seed. The additional run supports our previously reported results.
>
> > Q3. The code is not provided, however, and thus is hard to evaluate reproducibility, given the implementation of probabilistic models is by no means without certain complication.
>
> Though we provide a probabilistic interpretation of the DINO method, as stated above in Q1, we still use the same training algorithm as DINO. So, incorporating our interpretation does not add any additional complications to the implementation.

---

> > ### Comment · Reviewer_Xj5D · 2022-11-20
> > **Response to Rebuttal**
> >
> > Thank you the author(s) for the great effort responding to my reviews. After also learning from other reviews and responses, I would like to keep my score.

---

### Author Response · Authors · 2022-11-17
**General response to all reviewers**

We thank all the reviewers for the insightful comments and suggestions for improvements. The reason for our silence until now is that we have been implementing and running additional experiments, as suggested by reviewers Xj5D, SyZr and aACz. Specifically, to show the generality of our vMF interpretation of clustering-based SSL methods we have implemented a vMF version of iBOT [2] which attains new state-of-the-art results for the ViT-Base/16 architecture. Based on this, and other suggestions made by the reviewers, we have made the following changes to the paper:

* We emphasize that the vMF interpretation is not limited to DINO [1] and show results also for iBOT-vMF (kNN top-1 validation accuracy improved from 77.1% with iBOT to 78.66% with iBOT-vMF, linear classification accuracy improved from 79.5% with iBOT to 80.27% with iBOT-vMF). We believe that the paper title “DINO as a mixture of von-Mises Fisher components” is still suitable since, as we discuss in the paper, iBOT can be viewed as a derivative of DINO and, indeed, we still use DINO as our “running example” for discussing the proposed interpretation and modification.
* Minor updates to the abstract and introduction to mention about our newly obtained results when incorporating vMF in iBOT.
* In the appendix, we added a simple toy example involving only 2 clusters/mixture components to illustrate how DINO-vMF differs from DINO. We expand on the text in the first paragraph of section 3.3 where we discuss about when it is beneficial to increase the prototype magnitudes $\Vert \boldsymbol{w}^{(k)} \Vert$ for DINO-vMF and DINO when not $L^2$-normalizing the prototypes.
* DINO is known to be most effective when using Transformer backbones [1]. As suggested by reviewer SyZr we implemented the proposed method with another ViT architecture, specifically the ViT-Small/8 backbone that uses a smaller patch size of 8. Training this for 40 epochs, we observe that the results reflect our observations about the ViT-Small/16 backbone. The performance is on par or marginally better with DINO-vMF compared to DINO. Note that ViT-Small/8 pre-training already uses unnormalized prototypes. As we emphasize in the conclusion, we expect significant performance gains when comparing with a baseline that used $L^2$ normalized prototypes during pre-training. Though the performance gains are marginal with models that already use unnormalized prototypes, we show in section 5.2 that the learned vMF precision values have certain interpretable properties that are missing with the standard DINO prototypes.
* We conducted an additional run of the ablation studies using a different manually set seed. We report the average kNN classification accuracies over the 2 runs in the main paper and results of both runs are included in the appendix. Through this extra run, we strengthen the conclusions that we made from the ablation studies.
* We moved the main equations of our proposed DINO-vMF formulation (equations 8 and 9 in the old version of the paper, equations 6 and 7 in the newer version) to a more prominent position, just before section 3.4.
* To accommodate the iBOT evaluation results for the few-shot classification experiment in Table 3, we remove the variance values for the reported metric. Detailed results including the variance values are now available in the appendix.
* We moved the implementation details for the transfer task of linear classification on small datasets from the main paper to the appendix.
* We made minor changes to the notations used for vectors and column vectors.

**References**

[1] Mathilde Caron, Hugo Touvron, Ishan Misra, Hervé Jégou, Julien Mairal, Piotr Bojanowski, and Armand Joulin. Emerging properties in self-supervised vision transformers. In ICCV, 2021.

[2] Jinghao Zhou, Chen Wei, Huiyu Wang, Wei Shen, Cihang Xie, Alan Yuille, and Tao Kong. Image bert pre-training with online tokenizer. In ICLR, 2021.

---

### Decision · Program_Chairs · 2023-01-20

**Decision:**

Accept: notable-top-25%

**Justification For Why Not Higher Score:**

The paper proposed a modification to DINO (Self-distilled Visual Transformer) and the paper's novelty is not enough for an oral presentation.

**Justification For Why Not Lower Score:**

The results are solid and well-explained and it presents novel results on one important topic that is being extensively researched.

In any case, a presentation as a poster without a spotlight is also a fair outcome for this paper.

**Metareview: Summary, Strengths And Weaknesses:**

In this paper, the authors focus on the last layer of DINO and show it can be interpreted as a mixture model. The authors multiply the probabilities for the teacher and student networks by a normalizing Von Mises-Fischer constant to improve the network capacity. The reviewers are very positive about the results obtained by the proposed modifications and, the results are well-detailed and reproducible. The discussion between the authors and the reviewers has been very positive and has led to an improved paper.

**Note From Pc:**

if the above contains the word "oral" or "spotlight" please see: "oral" presentation means -> notable-top-5% and "spotlight" means -> notable-top-25%. As stated in our emails, we are disassociating presentation type from AC recommendations